# ContextVLA: Vision-Language-Action Model with Amortized Multi-Frame Context

## Abstract

Leveraging temporal context is crucial for success in partially observable robotic tasks. However, prior work in behavior cloning has demonstrated inconsistent performance gains when using multi-frame observations. In this paper, we introduce ContextVLA, a policy model that robustly improves robotic task performance by effectively leveraging multi-frame observations. Our approach is motivated by the key observation that Vision-Language-Action models (VLA), *i.e.*, policy models built upon a Vision-Language Model (VLM), more effectively utilize multi-frame observations for action generation. This suggests that VLMs' inherent temporal understanding capability enables them to extract more meaningful context from multi-frame observations. However, the high dimensionality of video inputs introduces significant computational overhead, making VLA training and inference inefficient. To address this, ContextVLA compresses past observations into a single context token, allowing the policy to efficiently leverage temporal context for action generation. Our experiments show that ContextVLA consistently improves over single-frame VLAs and achieves the benefits of full multi-frame training but with reduced training and inference times.

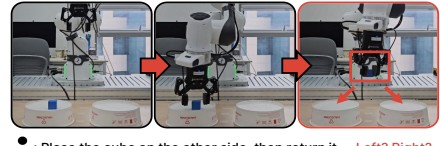
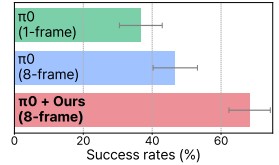
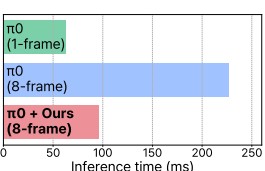

: Place the cube on the other side, then return it.    Left? Right?

(a) A task that requires temporal context  (b) Real-world task result  (c) Inference efficiency

Figure 1: **Overview.** (a) Many robotic tasks require temporal context to generate accurate actions. (b) By leveraging multi-frame observations, our proposed method, ContextVLA, achieves higher averaged success rates (%) over all baseline policies on real-world robotic tasks. (c) Moreover, our framework gets benefits of multi-frame training with reduced inference latency.

## 1 Introduction

Many robotic tasks are inherently non-Markovian, *i.e.*, the optimal decision at a given timestep $t$ cannot be determined from the latest observation $\mathbf{o}_t$ alone but requires past sequential observations $\mathbf{o}_{1:t}$ (Kaelbling et al., 1998; Zheng et al., 2024; Shi et al., 2025). For instance, an object may become occluded during manipulation (Shi et al., 2025). Solving long-horizon tasks may also require context about the previous motions of a robot, and handling dynamic environments often involves tracking the motion trajectories of moving objects (Zhang et al., 2025; Nasiriany et al., 2024). Consequently, policy models must have capability to predict the actions based on the understanding of consecutive input observations (*i.e.*, multi-frame observations) to perform real-world challenging task.

Despite its importance, recent behavior cloning (BC) policies usually have been trained with only a single frame observation (Kim et al., 2024; Bjorck et al., 2025; Pertsch et al., 2025; GEAR, 2025). This is mainly due to the mixed results reported in recent studies on training policy models with multi-frame observations. Specifically, several works argue that multi-frame observations do improve performance (Wu et al., 2023; Team et al., 2024; Cheang et al., 2024; Zheng et al., 2024; Liu et al., 2025; Li et al., 2025), but surprisingly, many others have observed contradictory results; namely, this training scheme can even lead to performance degradation (Muller et al., 2005; De Haan et al., 2019; Wen et al., 2020; Spencer et al., 2021; Seo et al., 2023; Torne et al., 2025).

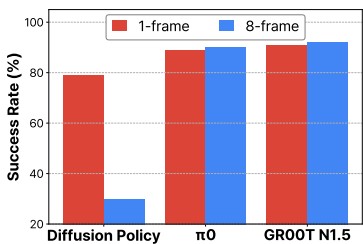 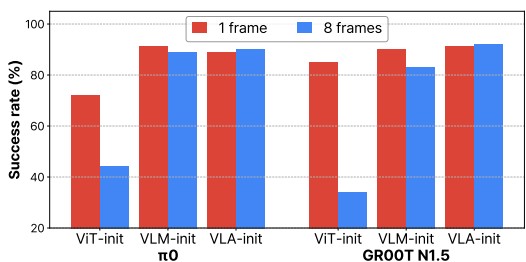

(a) Performance of recent policy models trained using either 1-frame or 8-frame observations.

(b) Performance of policy models of VLA architecture trained using either 1-frame or 8-frame observations under different weight initialization schemes.

Figure 2: **Effect of multi-frame observations for training various policy models.** We report the success rates (%) of various policy models fine-tuned on Square task from the Robomimic benchmark (Mandlekar et al., 2021). (a) When training policy models using multi-frame observations, traditional policy model (Diffusion policy) shows significant performance degradation, whereas recent Vision-Language-Action models (VLA; $\pi_0$ and GR00T N1.5) do not. (b) We find that the key factor in overcoming this problem is leveraging a pre-trained Vision-Language Model (VLM) to extract temporal information for action generation. ViT, VLM, and VLA-init indicate how the VLA architecture is initialized for training; we use a pre-trained vision encoder, VLM, or VLA, respectively, and other parameters are randomly initialized.

**Contribution.** We introduce a framework that enables BC policy models to effectively leverage multi-frame observations, thereby achieving consistent performance improvement across a wide range of manipulation tasks. To this end, we start by performing an analysis of why prior works have reported inconsistent gains from multi-frame observations. We find that while standard policies often suffer from performance degradation with multi-frame data, Vision-Language-Action models (VLA; Black et al. 2024; Bjorck et al. 2025), *i.e.*, policy models initialized from or conditioned on a Vision-Language Model (VLM; Beyer et al. 2024; Chen et al. 2025), mitigate this problem (Figure 2a). In particular, our analysis shows that the VLM serves as the key component in mitigating performance degradation (Figure 2b). This finding suggests that the VLM's temporal understanding is key to extracting more meaningful context from videos for action generation. However, leveraging multi-frame observations for VLA training and inference significantly increases computational cost, as high-dimensional sequences must be processed by large VLMs (often >1B parameters). Thus, it is important to develop efficient approaches for exploiting multi-frame information with VLAs.

Based on this analysis, we propose ContextVLA, an efficient framework that leverages a VLM's temporal understanding to learn a BC policy model that utilizes multi-frame observations. Our key idea is to compress past observations into a single context token, which enables the VLA to efficiently capture temporal context (*e.g.*, the movement of a robot) to generate actions with reduced computation overhead. Specifically, ContextVLA first processes the full observation sequence using the initial blocks of the VLM backbone. It then aggregates the tokens from past observations into a single context token. Then the remaining VLM blocks process the sequence consisting of this new context token and the tokens for the current observation. After that, the resulting VLM features are fed into an action decoder to generate actions in either an autoregressive (Pertsch et al., 2025) or diffusion-based manner (Black et al., 2024; Bjorck et al., 2025).

We verify the effectiveness of our scheme through extensive experiments on various robotic manipulation benchmarks, including Libero (Liu et al., 2023a), Simpler-WidowX (Li et al., 2024b), and Robocasa (Nasiriany et al., 2024). Our results show that ContextVLA consistently improves the performance of recent state-of-the-art VLAs that use single-frame observations. For instance, on the Simpler-WidowX benchmark, ContextVLA improves the average success rate of $\pi_0$ (Black et al., 2024) by 14.4% (41.8% → 56.2%). Moreover, we find that ContextVLA is particularly effective on long-horizon real-world robotic tasks that require temporal understanding (Figure 4b). For example, fine-tuning $\pi_0$ (Black et al., 2024) with our framework achieves a 65% success rate on the pick-and-place twice (PnP Twice) task, whereas the single-frame baseline gets only 25%. Finally, we show that ContextVLA enables efficient training and inference of multi-frame VLA models, delivering the benefits of multi-frame training while significantly reducing the training and inference costs.

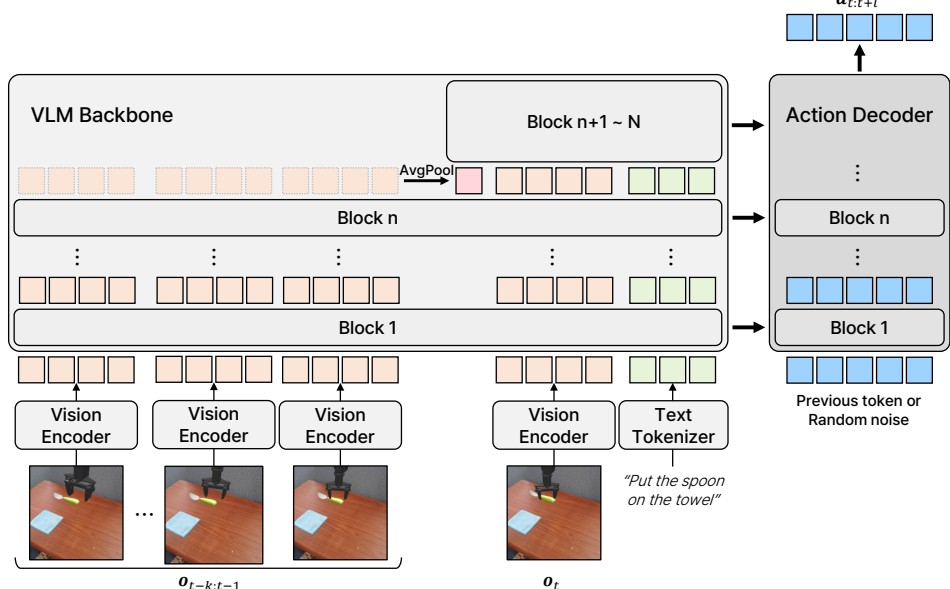

Figure 3: **Overview of ContextVLA.** We design an efficient Vision-Language-Action model (VLA) that generates actions using multi-frame visual observations. We use a Vision-Language Model (VLM) to encode observations $\mathbf{o}_{t-k:t}$, where we compress past observations $\mathbf{o}_{t-k:t-1}$ into a single context token $\mathbf{m}$ at the VLM block $n$. We then leverage the VLM features to generate actions via either autoregressive modeling or diffusion-based modeling.

## 2 METHOD

In this section, we introduce ContextVLA, an efficient training framework for Vision-Language-Action models (VLA) that leverages *multi-frame* observations. In a nutshell, ContextVLA encodes hidden states of past visual observations into a compact global context token at the intermediate layer of Vision-Language-Model (VLM) backbone, while preserving the number of tokens of current observations. After that, these VLM features are fed into an action decoder to generate the action. We provide the overview of ContextVLA in Figure 3.

### 2.1 PRELIMINARIES

**Problem Setup.** Let $\tau = \{(\mathbf{o}_t, \mathbf{c}_t, \mathbf{a}_t)\}_{t=1}^T$ be an expert demonstration consisting of visual observation $\mathbf{o}_t$, language instruction $\mathbf{c}_t$, and robot action $\mathbf{a}_t$ for each timestep $t$, and let $\mathcal{D} = \{\tau_i\}_{i=1}^N$ be a dataset consisting of expert demonstrations. Here, visual observation $\mathbf{o}_t$ can be either single or multi-view depending on the environment. Moreover, we denote by $\mathbf{x}_{a:b}$ the sequence of consecutive vectors $[\mathbf{x}_a, \ldots, \mathbf{x}_b]$ for $a < b$.

Given $\tau \in \mathcal{D}$, our goal is to train a policy model $\pi_\theta(\mathbf{a}_{t:t+l}|\mathbf{o}_{t-k:t}, \mathbf{c}_t)$, which predicts $l + 1$ future actions $\mathbf{a}_{t:t+l}$ of the robot (Zhao et al., 2023; Chi et al., 2023) from $k + 1$-frame observations $\mathbf{o}_{t-k:t}$ and a language instruction $\mathbf{c}_t$. In particular, we aim this policy $\pi_\theta$ to leverage $k + 1$-frame visual observations so that it understands not only the current state, but the context of past observations to generate actions. However, using longer past observations can dramatically increase computation and memory overheads by increasing input dimensionality that $\pi_\theta$ processes, resulting in inefficient training and inference. Therefore, we design $\pi_\theta$ to process past observations efficiently.

**Multi-frame Vision-Language-Action Model.** We design the policy $\pi_\theta$ as VLA, where it encodes visual observations $\mathbf{o}_{t-k:t}$ and a task instruction $\mathbf{c}_t$ using a pre-trained VLM (Beyer et al., 2024; Bai et al., 2025; Chen et al., 2025), and uses the extracted features from the VLM to generate a robot action $\mathbf{a}_{t:t+l}$. Action generation can be performed in either an autoregressive (Kim et al., 2024; Pertsch et al., 2025) or a diffusion-based manner (Black et al., 2024; Bjorck et al., 2025), conditioning on the VLM features. We describe the detailed action generation process in Appendix F.

## 2.2 CONTEXTVLA

To leverage past observations for action generation, ContextVLA uses Vision-Language Model (VLM) to process multi-frame observations. However, video inputs contain many tokens, substantially increasing compute and memory overhead in the VLM. To address this, our key idea is to compress past observations into a single context token, allowing the VLM to capture the temporal context of past observations, *e.g.*, movement of the robot, while reducing computation and memory overhead. Specifically, we compress the past observations at the intermediate layer of the VLM backbone by applying average pooling. And then, an action decoder generates robot actions conditioned on the resulting VLM features.

Formally, given multi-frame observations $\mathbf{o}_{t-k:t}$, we first process them all through a vision encoder $f$ to obtain visual features $\mathbf{e}_{t-k:t} = f(\mathbf{o}_{t-k:t})$. We then use $\mathbf{x} = [\mathbf{e}_{t-k:t}, \mathbf{c}_t]$ *i.e.*, a concatenation of $\mathbf{e}_{t-k:t}$ and $\mathbf{c}_t$, as input tokens to the VLM backbone $g$.

**Amortization of Past Observation.** The next step is to process the input tokens $\mathbf{x}$ with the VLM backbone $g$ to extract the features that will be used to generate the action. To efficiently process multi-frame observations using the VLM backbone, we compress past observations into a single context token within the intermediate layer of the VLM model $g$.

Formally, let $N$ be the number of VLM backbone blocks. We split the VLM backbone into two parts at the $n$-th block. First, in the first $n$ blocks, we process all tokens of $\mathbf{x}$ through the VLM blocks to get intermediate hidden states $\mathbf{h} = [\mathbf{h}_{t-k:t}, \mathbf{h_c}]$. In particular, when tokens are fed into the self-attention layers, we apply a causal mask to the visual tokens, regardless of the original VLM attention mask. This allows efficient inference by processing and caching past observations before the next timestep. After this, we compress the past visual observations into a context token using average pooling, *i.e.*, $\mathbf{m} = \texttt{AvgPool}([\mathbf{h}_{t-k:t-1}])$ to capture temporal context of past observations. In the remaining $N - n$ blocks, we replace the hidden states of the past observations $\mathbf{h}_{t-k:t-1}$ with the context token $\mathbf{m}$, and process $[\mathbf{m}, \mathbf{h}_t, \mathbf{h_c}]$ through the blocks. As a result, we obtain the VLM features, which encode both the current observation and an amortized context of past observations.

**Action Decoder.** Our action decoder generates a chunk of robot action $\mathbf{a}_{t:t+l}$ with a length $l + 1$ (Zhao et al., 2023; Chi et al., 2023), using VLM features as conditioning. Because our amortization scheme does not depend on the type of action decoders, ContextVLA can be applied to any VLAs regardless of their action decoder models. Specifically, it can be applied to autoregressive (Kim et al., 2024; Bu et al., 2025b; Pertsch et al., 2025) or diffusion-based action decoders (Black et al., 2024; Bjorck et al., 2025; GEAR, 2025). For instance, for the autoregressive modeling, we encode an action $\mathbf{a}_{t:t+l}$ into discrete action tokens using the action tokenizer (Bu et al., 2025b; Pertsch et al., 2025), and then use the same VLM backbone to generate the discrete action tokens via next action token prediction. For the diffusion-based modeling, we generate an action using a diffusion transformer (DiT Peebles & Xie (2022)) conditioning on the VLM hidden states, *e.g.*, output tokens of the VLM or key-values in the VLM blocks.

**Training Objective.** We train ContextVLA to predict the target ground-truth action $\mathbf{a}_{t:t+l}$ in the expert trajectory $\tau = \{(\mathbf{o}_t, \mathbf{c}_t, \mathbf{a}_t)\}_{t=1}^T$. Specifically, we train the model $\pi_\theta$ to minimize the loss $\ell(\pi_\theta(\mathbf{o}_{t-k:t}, \mathbf{c}_t), \mathbf{a}_{t:t+l})$, where $\ell$ corresponds to either a next-token prediction loss for the autoregressive action modeling or a flow-matching loss for the diffusion-based action modeling.

**Efficient Inference via KV-caching.** ContextVLA generates actions faster than a VLA that uses videos without compression, since we use an amortized token instead of past observations in most VLM blocks. In addition to this, we further make an inference of ContextVLA faster by processing as many of the observations as possible before the next timestep. Specifically, at timestep $t - 1$, we process $\mathbf{o}_{t-k:t-1}$ through the first $n$ VLM blocks to obtain a KV-cache for each block, and obtain a context token $\mathbf{m}$. At timestep $t$, since we explicitly implement the attention layers of the first $n$ VLM blocks with causal-attention, we generate actions using $\mathbf{o}_t$, pre-computed $\mathbf{m}$, and the KV-cache, rather than re-processing $\mathbf{o}_{t-k:t-1}$.

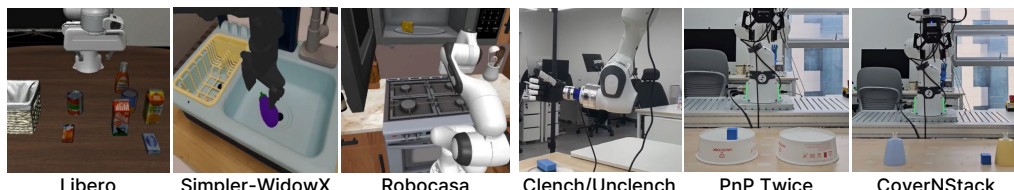

| Libero | Simpler-WidowX | Robocasa | Clench/Unclench | PnP Twice | CoverNStack |

(a) Simulated robotic manipulation tasks      (b) Real-world robotic tasks

Figure 4: **Examples of visual observations from the evaluation tasks.** (a) We consider simulated robotic manipulation tasks from Libero (Liu et al., 2023a), Simpler-WidowX (Li et al., 2024b), and Robocasa (Nasiriany et al., 2024). (b) We design real-world robotic tasks: Clench/unclench hand (Clench/Unclench), pick-and-place twice (PnP Twice), and cover and stack (CoverNStack).

## 3 EXPERIMENTS

We design experiments to investigate the following questions:

- Can ContextVLA leverage the multi-frame observations to perform diverse robotic tasks without performance degradation? (Tables 1 to 3) In particular, is ContextVLA effective on robotic tasks that particularly need past observations? (Table 4)
- Is ContextVLA efficient during training and inference? (Table 5 and Figure 5)
- What is the effect of each component of ContextVLA? (Table 6)

### 3.1 EXPERIMENTAL SETUP

**Implementation Details.** We report the performance of our method, ContextVLA, by fine-tuning pre-trained Vision-Language-Action models (VLA). Specifically, we use $\pi_0$ (Black et al., 2024), GR00T N1.5 (GEAR, 2025), and $\pi_0$-FAST (Pertsch et al., 2025) by following their official implementation. For $\pi_0$ and $\pi_0$-FAST, we perform full-finetuning, *i.e.*, we update all model parameters, but we freeze the vision encoder and VLM backbone for GR00T N1.5. We use 8 consecutive frames as multi-frame observations, and we compress past observations into context tokens at the output of the 2nd VLM block (*i.e.*, $n$=2). We report the best performance by evaluating the models at fixed intervals during training. We provide more detailed implementation details in Appendix A.

**Pre-training Details.** We also report the performance of ContextVLA by fine-tuning ContextVLA pre-trained on the OXE Magic Soup dataset (O'Neill et al., 2024; Kim et al., 2024). We initialize the model from Qwen2.5VL 3B (Bai et al., 2025) and pre-train it to generate discrete action tokens using the FAST tokenizer (Pertsch et al., 2025). We pre-train our model for 180K iterations, using the AdamW optimizer with a batch size of 128.

**Baselines.** We compare the performance of ContextVLA with recent open-source VLAs: Octo (Team et al., 2024), OpenVLA (O'Neill et al., 2024), RoboVLMs (Liu et al., 2025), TraceVLA (Zheng et al., 2024), SpatialVLA (Qu et al., 2025), NORA (Hung et al., 2025), $\pi_0$ (Black et al., 2024), GR00T N1.5 (GEAR, 2025), and $\pi_0$-FAST (Pertsch et al., 2025). In particular, to evaluate the benefit of ContextVLA, we compare $\pi_0$, GR00T N1.5, and $\pi_0$-FAST trained and fine-tuned with single-frame inputs against their counterparts fine-tuned with multi-frame inputs using our method. All models are fine-tuned with the same batch size and number of iterations, and we report their best success rates at fixed evaluation intervals. We describe more details of baselines in Appendix C.

### 3.2 SIMULATED ROBOTIC TASKS

To demonstrate our method can leverage multi-frame visual observations, we first evaluate our method by fine-tuning pre-trained VLAs on diverse simulated robotic manipulation benchmarks (see Figure 4a for examples of tasks from the benchmarks used in our experiments).

Table 1: **Results on Libero.** We report the success rates (%) of various VLAs fine-tuned on the training dataset of Libero (Liu et al., 2023a; Kim et al., 2024). For $\pi_0$ (Black et al., 2024), $\pi_0$-FAST (Pertsch et al., 2025), and GR00T N1.5 (GEAR, 2025), we report the performance with a standard deviation of 3 random training seeds by fine-tuning the pre-trained model with the official implementations, and other reported numbers borrow from Team et al. (2024); Hung et al. (2025).

| Method | # frames | Spatial | Object | Goal | Long | Avg. |
|---|---|---|---|---|---|---|
| *VLAs pretrained on OXE dataset (O'Neill et al., 2024)* | | | | | | |
| Octo (Team et al., 2024) | 2 | 78.9 | 85.7 | 84.6 | 51.1 | 75.1 |
| OpenVLA (Kim et al., 2024) | 1 | 84.9 | 88.4 | 79.2 | 53.7 | 76.5 |
| TraceVLA (Zheng et al., 2024) | 6 | 84.9 | 85.2 | 75.1 | 54.1 | 74.8 |
| SpatialVLA (Qu et al., 2025) | 1 | 88.2 | 89.9 | 78.6 | 55.5 | 78.1 |
| NORA (Hung et al., 2025) | 1 | 92.2 | 95.4 | 89.4 | 74.6 | 87.9 |
| **ContextVLA (Ours)** | 8 | **95.8** | **99.2** | **92.6** | **87.0** | **93.7** |
| *VLAs pretrained on external datasets* | | | | | | |
| $\pi_0$ (Black et al., 2024) | 1 | $96.3_{\pm0.3}$ | $97.3_{\pm0.4}$ | $96.2_{\pm0.3}$ | $88.8_{\pm0.3}$ | $94.7_{\pm0.1}$ |
| **+ ContextVLA (Ours)** | 8 | $\mathbf{97.9}_{\pm0.5}$ | $\mathbf{98.9}_{\pm0.6}$ | $\mathbf{96.3}_{\pm0.3}$ | $\mathbf{93.1}_{\pm0.6}$ | $\mathbf{96.6}_{\pm0.1}$ |
| $\pi_0$-FAST (Pertsch et al., 2025) | 1 | $96.3_{\pm1.1}$ | $97.5_{\pm0.9}$ | $94.5_{\pm0.6}$ | $84.8_{\pm1.4}$ | $93.3_{\pm0.5}$ |
| **+ ContextVLA (Ours)** | 8 | $\mathbf{97.8}_{\pm0.6}$ | $\mathbf{98.9}_{\pm0.4}$ | $\mathbf{95.9}_{\pm0.6}$ | $\mathbf{90.8}_{\pm0.9}$ | $\mathbf{95.8}_{\pm0.2}$ |
| GR00T N1.5 (GEAR, 2025) | 1 | $98.0_{\pm0.5}$ | $\mathbf{99.3}_{\pm0.2}$ | $96.9_{\pm0.3}$ | $88.7_{\pm0.8}$ | $95.7_{\pm0.2}$ |
| **+ ContextVLA (Ours)** | 8 | $\mathbf{98.6}_{\pm0.2}$ | $99.1_{\pm0.2}$ | $\mathbf{97.3}_{\pm0.1}$ | $\mathbf{93.0}_{\pm0.3}$ | $\mathbf{97.0}_{\pm0.1}$ |

Table 2: **Results on Simpler-WidowX.** We report the success rates (%) of various VLAs fine-tuned on the Bridgev2 dataset (Walke et al., 2023). For $\pi_0$ (Black et al., 2024), $\pi_0$-FAST (Pertsch et al., 2025), and GR00T N1.5 (GEAR, 2025), we report the performance with a standard deviation of 3 random training seeds by fine-tuning the pre-trained model with the official implementations, and other reported numbers borrow from Qu et al. (2025).

| Method | # frames | Spoon on Towel | Carrot on Plate | Stack Cube | Put Eggplant in Basket | Avg. |
|---|---|---|---|---|---|---|
| *VLAs pretrained on OXE dataset (O'Neill et al., 2024)* | | | | | | |
| Octo-base (Team et al., 2024) | 2 | 12.5 | 8.3 | 0.0 | 43.1 | 16.0 |
| Octo-small (Team et al., 2024) | 2 | 47.2 | 9.7 | 4.2 | 56.9 | 29.5 |
| OpenVLA (Kim et al., 2024) | 1 | 0.0 | 0.0 | 0.0 | 4.1 | 1.0 |
| RoboVLMs (Liu et al., 2025) | 16 | 29.2 | 25.0 | 12.5 | 58.3 | 31.3 |
| SpatialVLA (Qu et al., 2025) | 1 | 16.7 | 25.0 | 29.2 | **100.0** | 42.7 |
| **ContextVLA (Ours)** | 8 | **52.0** | **56.0** | **58.0** | 72.0 | **59.5** |
| *VLAs pretrained on external datasets* | | | | | | |
| $\pi_0$ (Black et al., 2024) | 1 | $46.7_{\pm3.3}$ | $38.7_{\pm7.1}$ | $\mathbf{42.7}_{\pm3.3}$ | $39.3_{\pm8.4}$ | $41.8_{\pm3.2}$ |
| **+ ContextVLA (Ours)** | 8 | $\mathbf{53.3}_{\pm1.8}$ | $\mathbf{56.0}_{\pm2.3}$ | $41.3_{\pm2.7}$ | $\mathbf{74.0}_{\pm7.2}$ | $\mathbf{56.2}_{\pm1.8}$ |
| $\pi_0$-FAST (Pertsch et al., 2025) | 1 | $59.0_{\pm1.2}$ | $79.0_{\pm1.2}$ | $65.0_{\pm1.9}$ | $33.0_{\pm1.0}$ | $59.0_{\pm0.7}$ |
| **+ ContextVLA (Ours)** | 8 | $\mathbf{60.7}_{\pm1.5}$ | $\mathbf{81.3}_{\pm5.4}$ | $\mathbf{78.7}_{\pm4.3}$ | $\mathbf{62.0}_{\pm2.9}$ | $\mathbf{70.7}_{\pm1.7}$ |
| GR00T N1.5 (GEAR, 2025) | 1 | $\mathbf{30.0}_{\pm1.2}$ | $28.0_{\pm1.2}$ | $\mathbf{16.0}_{\pm3.5}$ | $42.7_{\pm1.8}$ | $29.2_{\pm0.9}$ |
| **+ ContextVLA (Ours)** | 8 | $28.0_{\pm2.3}$ | $\mathbf{29.3}_{\pm0.7}$ | $14.7_{\pm3.3}$ | $\mathbf{50.3}_{\pm2.4}$ | $\mathbf{31.8}_{\pm1.5}$ |

**Experimental Setup.** We first consider Libero benchmark (Liu et al., 2023a), one of the popular benchmarks for evaluating VLA, which includes 4 different sub-benchmarks (Spatial, Object, Goal, and Long) that contain 10 tasks each. We also consider 4 tasks from the Simpler-WidowX benchmark (Li et al., 2024b), which is a more challenging setup due to a visual gap between real-world training data and simulated test environments (Real-to-Sim). We report the performance by fine-tuning the models on Bridge v2 dataset (Ebert et al., 2021). Moreover, we consider the Robocasa benchmark (Nasiriany et al., 2024), which includes 24 tasks in simulated kitchen environments. It consists of more than 2500 different kitchen scenes across more than 150 object categories, requiring a policy to have instruction following and generalization ability over the scene and object. We report the performance by fine-tuning all models on the machine-generated training dataset, consisting of 100 demos per task. We provide a more detailed experimental setup in Appendix A and an explanation about the benchmark with evaluation details in Appendix B.1.

Table 3: **Results on Robocasa.** We report the success rates (%) of various VLAs fine-tuned on the training dataset of Robocasa (Nasiriany et al., 2024), consisting of 24 tasks with 100 demos per task. We report the performance with a standard deviation of 3 random training seeds by fine-tuning the pre-trained model with the official implementations.

| Method | # frames | Pick and Place | Others | Avg. |
|---|---|---|---|---|
| $\pi_0$ (Black et al., 2024) | 1 | $32.9_{\pm 0.4}$ | $70.3_{\pm 1.4}$ | $57.9_{\pm 0.9}$ |
| **+ ContextVLA (Ours)** | 8 | $\mathbf{35.6}_{\pm 0.8}$ | $\mathbf{70.4}_{\pm 0.2}$ | $\mathbf{58.8}_{\pm 0.2}$ |
| $\pi_0$-FAST (Pertsch et al., 2025) | 1 | $46.1_{\pm 1.0}$ | $68.1_{\pm 0.8}$ | $60.3_{\pm 0.1}$ |
| **+ ContextVLA (Ours)** | 8 | $\mathbf{48.6}_{\pm 0.6}$ | $\mathbf{68.7}_{\pm 0.6}$ | $\mathbf{62.0}_{\pm 0.2}$ |
| GR00T N1.5 (GEAR, 2025) | 1 | $51.8_{\pm 1.4}$ | $67.6_{\pm 1.1}$ | $62.3_{\pm 0.3}$ |
| **+ ContextVLA (Ours)** | 8 | $\mathbf{52.8}_{\pm 0.5}$ | $\mathbf{70.2}_{\pm 0.5}$ | $\mathbf{64.4}_{\pm 0.2}$ |

Table 4: **Results on real-world robotic tasks.** We report the success rates (%) of various VLAs fine-tuned on the training dataset of each task. We report the performance by fine-tuning the pre-trained model with the official implementations.

| Method | # frames | Clench/Unclench | PnP Twice | | CoverNStack | |
|---|---|---|---|---|---|---|
| | | | PnP Once | Full | Cover Cube | Full |
| $\pi_0$ (Black et al., 2024) | 1 | 40.0 | 55.0 | 25.0 | 60.0 | 45.0 |
| $\pi_0$ (Black et al., 2024) | 8 | 40.0 | 60.0 | 55.0 | 65.0 | 45.0 |
| **+ ContextVLA (Ours)** | 8 | **80.0** | **75.0** | **65.0** | **85.0** | **60.0** |
| GR00T N1.5 (Bjorck et al., 2025) | 1 | 20.0 | 55.0 | 15.0 | 50.0 | 10.0 |
| GR00T N1.5 (Bjorck et al., 2025) | 8 | **80.0** | 60.0 | 30.0 | 50.0 | 25.0 |
| **+ ContextVLA (Ours)** | 8 | **80.0** | **70.0** | **50.0** | **55.0** | **35.0** |

**Results.** We find that our framework consistently improves the baselines as shown in Tables 1 to 3. This demonstrates that our method indeed can leverage multi-frame visual observations to generate action. Specifically, we first find that ContextVLA, pretrained on the OXE dataset, outperforms the video-based VLAs, demonstrating the effectiveness of ContextVLA over other multi-frame-based approaches (see Tables 1 and 2). We also observe that ContextVLA improves the baselines by a significant margin in Simpler-WidowX Benchmark (Table 2), a challenging setup due to the Real-to-Sim gap. For instance, ContextVLA improves the averaged success rates of $\pi_0$ by 14.6% (41.8% $\rightarrow$ 56.2%). In addition, our framework improves the performance in the Robocasa Benchmark (Table 3). In particular, we observe that our framework improves the performance in Pick and Place (PnP) tasks, consisting of diverse target objects and positions. For instance, ContextVLA improves the averaged success rates of $\pi_0$ in PnP by 2.7% (32.9% $\rightarrow$ 35.6%). This demonstrates that the performance gains with ContextVLA are not specific to the training setup, but instead highlight its ability to generalize across diverse objects and locations.

### 3.3 REAL-WORLD ROBOTIC TASKS

To investigate whether our method can leverage multi-frame visual observations, we further evaluate our method on real-world robotic tasks that require temporal context to perform the task successfully (see Figure 4b for examples of tasks used in our experiments).

**Tasks.** We design several real-world robotic tasks as follows (see Appendix B.2 for more details):

- **Clench/Unclench.** The policy should clench and unclench hand of the humanoid robot repeatedly. At an intermediate state between clenching and unclenching, it must decide to grasp or release the hand depending on the previous movement.
- **PnP Twice.** Given a cube and two plates (A and B), the policy should move the cube from plate A to B and then back from B to A. At each step, it must decide (a) whether to close the gripper (pick) or open it (place), and (b) which plate to place the cube on, based on the previous action.
- **CoverNStack.** Given a cube and two cups, the policy should cover the cube with the closest cup, and then stack the other cup on top of the covered cup. After covering the object with a cup, it must decide which cup to grasp and place it on top of the other cup, depending on the last movement.

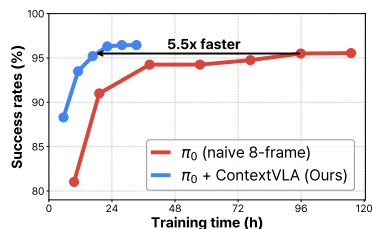

Figure 5: **Training efficiency.** We report the wall clock time of fine-tuning $\pi_0$ on Libero (Liu et al., 2023a) using 4 NVIDIA A100 80GB GPU.

Table 6: **Component-wise analysis** on Libero (Liu et al., 2023a) and Simpler-WidowX (Li et al., 2024b) benchmarks. All models are $\pi_0$ (Black et al., 2024) fine-tuned for 60K iterations with a batch size of 32, using the Libero training dataset for Libero and the BridgeV2 dataset (Ebert et al., 2021) for Simpler-WidowX. We report the averaged success rates (%) of the models on each benchmark.

Table 5: **Inference efficiency.** We report inference time (ms) required for $\pi_0$ (Black et al., 2024) to generate action from 8-frame, 2-view observations using a single NVIDIA A100 80GB GPU.

| Compression | KV-caching | Time (ms) |
|:---:|:---:|:---:|
| ✗ | - | 227.2 |
| ✓ | ✗ | 129.9 |
| ✓ | ✓ | **96.3** |

| # frames | Depth | Context Token | Libero | Simpler-WidowX |
|:---:|:---:|:---:|:---:|:---:|
| 1 | $\pi_0$ (Black et al., 2024) | | 94.6 | 41.8 |
| 8 | $\pi_0$ (Black et al., 2024) | | 95.6 | 47.8 |
| 2 | 2 | ✓ | 94.8 | 50.5 |
| 4 | 2 | ✓ | 95.0 | 52.0 |
| 8 | 2 | ✓ | **96.5** | **56.2** |
| 8 | 1 | ✓ | 96.1 | 46.5 |
| 8 | 2 | ✓ | **96.5** | **56.2** |
| 8 | 4 | ✓ | 96.4 | 53.5 |
| 8 | 6 | ✓ | 95.8 | 51.5 |
| 8 | 8 | ✓ | 96.4 | 48.5 |
| 8 | 2 | ✗ | 95.6 | 49.0 |
| 8 | 2 | ✓ | **96.5** | **56.2** |

**Training and Evaluation Setup.** For each task, we collect 50 demonstrations and report the performance by fine-tuning $\pi_0$ (Black et al., 2024) and GR00T N1.5 (GEAR, 2025) using our framework on the collected demonstrations. We train all models for 30K iterations using the AdamW optimizer with a batch size of 32. We evaluate each model with 20 trials per task and report partial (performs single time for repetitive tasks, and completes only the first subtask for long-horizon tasks) and full success rates. We describe the details of the evaluation in Appendix B.2.

**Results.** In Table 4, we find that ContextVLA improves the performance of the baselines by a significant margin. While single-frame baselines often fail to perform simple repetitive actions, *e.g.,* clench/unclench, ContextVLA consistently outperforms them. This indicates that ContextVLA leverages temporal context to resolve temporal ambiguities. We observe the same results in sequential tasks (PnP Twice and CoverNStack), *e.g.,* ContextVLA achieves the highest success rates on both tasks, but single-frame baselines often succeed partially. This further demonstrates the effectiveness of ContextVLA on long-horizon tasks that require temporal understanding. In particular, our framework even outperforms VLAs that utilize 8-frame observations without compression. For instance, in the CoverNStack tasks, fine-tuning $\pi_0$ with ContextVLA gets a 60% success rate, whereas fine-tuning it with 8-frame observations without compression gets 45%. The improvement can be attributed to the faster inference speed of our framework, as latency is known to degrade performance during real-world robot deployment (Black et al., 2025). This highlights the benefit of our approach that compresses the past observations. We provide the qualitative results in Appendix I.

### 3.4 ANALYSIS AND ABLATION STUDY

We first perform efficiency analysis in Figure 5 and Table 5. After that, in Table 6, we provide ablation studies to investigate the effect of each component of ContextVLA. We include more analysis about the amortized context tokens in the Appendix E.

**Training Efficiency.** In Figure 5, we analyze how efficient our compression scheme makes VLA when training using multi-frame observations. We compare the success rates under the same training wall-clock time with $\pi_0$ that uses 8-frame visual observations. We find that our framework is much faster to achieve the best performance of the $\pi_0$, *e.g.*, 5.5× faster on the Libero dataset.

**Inference Efficiency.** In Table 5, we measure the inference time of our framework to evaluate the inference efficiency of our compression scheme with multi-frame observations. We find that our framework is 2.4× faster than $\pi_0$ with 8-frame visual observations without compression. In particular, the efficient inference scheme with KV-caching reduces latency by 33.6 ms.

**Number of Past Observations.**  We investigate the scalability of ContextVLA with the number of past observations. We evaluate this by fine-tuning $\pi_0$ (Black et al., 2024) with ContextVLA using different numbers of frames from 2 to 8. We find that the success rates consistently increase as the number of past observations increases. For instance, using 4 frames achieves a success rate of 52.0% on the Simpler-WidowX benchmark, while using 8 frames achieves a success rate of 56.2%.

**Depth for Token Compression.**  We also investigate the appropriate depth $n$ for compressing past observations. By fine-tuning $\pi_0$ with past observations compressed into a context token at different backbone depths, we find that compression at shallow blocks ($n = 2$) is the optimal choice, while compression at other depths still shows the performance improvement. We note that the compression at shallow block allows the model to enhance efficiency during training and inference (see Table 5 and Figure 5).

**Effect of Amortized Context Token.**  We further investigate whether the compressed context token indeed provides meaningful information to generate actions. To evaluate this, we fine-tune $\pi_0$ by compressing past observations at a middle block and compare two variants: using the compressed tokens in the remaining blocks or discarding them. We find that using context token improves the performance by a large margin, *e.g.*, in Simpler-WidowX, 49.0% → 56.2%. This demonstrates that the context token captures temporal context from past observations, enabling the policy to generate actions better. We also find that $\pi_0$ using context token even outperforms $\pi_0$ trained on 8-frame observations without compression. We hypothesize this is because the context token summarizes past observations and mitigates redundancy across consecutive frames.

## 4 RELATED WORK

**Vision-Language-Action Models (VLA).**  Recently, VLAs (O'Neill et al., 2024; Zitkovich et al., 2023; Kim et al., 2024) have emerged as a promising framework for general robot policy that can perform many different tasks with a single model, by leveraging pre-trained Vision-Language Model (VLM) (Liu et al., 2023b; Beyer et al., 2024; Bai et al., 2025; Chen et al., 2025) and training on large-scale robot manipulation datasets (Ebert et al., 2021; Walke et al., 2023; O'Neill et al., 2024; Khazatsky et al., 2024; Bu et al., 2025a). However, many existing VLAs are trained to process only a single-frame visual observation to generate actions (Kim et al., 2024; Li et al., 2024a; Black et al., 2024; Shukor et al., 2025; Yang et al., 2025; Bjorck et al., 2025; Hung et al., 2025; Qu et al., 2025; Driess et al., 2025; GEAR, 2025; Cheang et al., 2025), limiting their ability to perform diverse robotic tasks that require temporal context. To address this, several recent approaches have attempted to leverage multi-frame observations. A common strategy is to use full context of multi-frame observations to generate actions (Wu et al., 2023; Team et al., 2024; Cheang et al., 2024; Huang et al., 2025; Liu et al., 2025; Wang et al., 2025). However, using multi-frame observations without compression increases computation and memory overheads by increasing input dimensionality that VLA processes, resulting in inefficient training and inference. In contrast, a recent approach, TraceVLA (Zheng et al., 2024), summarizes the observations by tracking the robot trace. However, it requires external point tracking model (Karaev et al., 2024), making inference still slow. In this paper, we introduce an efficient VLA framework that leverages multi-frame visual observations.

**Efficient Training and Inference of Multi-frame Policy.**  A higher input dimensionality compared to single-frame observations makes the training and inference of policy model inefficient. This hinders scaling up the policy model training (Black et al., 2024; Bjorck et al., 2025), and poses a significant challenge for deployment, as inference speed is a critical issue in robotics (Black et al., 2025). To address this, some approaches handle multi-frame inputs efficiently by compressing past observations (Wen et al., 2020; Seo et al., 2023), selecting a key-frames (Wen et al., 2021), or summarizing entire sequences into high-level visualizations (Sundaresan et al., 2024; Zheng et al., 2024). In addition, recent work proposes two-stage scheme where it first trains a single-frame policy, and then trains a multi-frame policy after freezing visual encoder (Torne et al., 2025). Our work also proposes a method that allows the policy to leverage multi-frame observation more efficiently that compressing past observations into a single token.

## 5 CONCLUSION

In this work, we have presented ContextVLA, an efficient framework for Vision-Language-Action models (VLA) that leverages multi-frame visual observations for action generation. Motivated by the observation that VLAs mitigate the performance degradation suffered by behavior cloning policies with multi-frame inputs, we introduce a simple yet effective method that compresses past observations into a single context token, allowing the VLA to capture temporal context more efficiently. Our experiments show that ContextVLA leverages multi-frame observations to improve the performance of existing VLAs. We also find that our scheme retains the benefits of multi-frame training with less training and inference time. We hope that our work facilitates future research toward generalist robot policies that can capture temporal context to perform more diverse tasks.

## REPRODUCIBILITY STATEMENT

For the reproducibility of our results, we provide the implementation details in Appendices A and B, including training and inference setups. In addition, we will open-source the source code with the model checkpoint.

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

# A  IMPLEMENTATION DETAILS

We report the performance of ContextVLA, by fine-tuning pre-trained Vision-Language-Action models (VLA). Specifically, we use $\pi_0$ (Black et al., 2024), GR00T N1.5 (Bjorck et al., 2025), and $\pi_0$-FAST (Pertsch et al., 2025) by following their official implementation: For $\pi_0$ and $\pi_0$-FAST, we fine-tune all model parameters, and for GR00T N1.5, we freezes vision encoder and VLM backbones. We use 8 consecutive frames as multi-frame observations, and we compress past observations into context tokens at the output of the 2nd VLM block (*i.e.*, $n$=2). For simulated tasks, we train $\pi_0$ + ContextVLA, GR00T N1.5 + ContextVLA for 60K iterations, but train $\pi_0$-FAST + ContextVLA for 30K iterations as $\pi_0$-FAST converges faster than $\pi_0$ and GR00T N1.5 (Pertsch et al., 2025). For real-world tasks, we only need to train the model on a single task with 50 demonstrations, so we train $\pi_0$ + ContextVLA and GR00T N1.5 + ContextVLA for fewer (30K) iterations.

Table 7: Hyperparameter details of training ContextVLA in simulated robotic tasks

|  | $\pi_0$ + ContextVLA | $\pi_0$-FAST + ContextVLA | GR00T N1.5 + ContextVLA |
|---|---|---|---|
| optimizer | AdamW | AdamW | AdamW |
| optimizer momentum | $\beta_1, \beta_2 = 0.9, 0.95$ | $\beta_1, \beta_2 = 0.9, 0.95$ | $\beta_1, \beta_2 = 0.95, 0.999$ |
| optimizer weight decay | 1e-10 | 1e-10 | 1e-5 |
| learning rate | 2.5e-5 | 2.5e-5 | 1e-4 |
| learning rate scheduler | Cosine decay | Cosine decay | Cosine decay |
| warmup iterations | 1000 | 1000 | 3000 |
| batch size | 32 | 32 | 32 |
| training iterations (simulated tasks) | 60000 | 30000 | 60000 |
| training iterations (real-world tasks) | 30000 | - | 30000 |

# B  BENCHMARK DETAILS

## B.1  SIMULATED ROBOTIC MANIPULATION BENCHMARKS

We evaluate our method on the following simulated robotic manipulation benchmarks.

**Libero.**  We consider Libero benchmark (Liu et al., 2023a), a widely used simulated robotic manipulation benchmark for evaluating the performance of Vision-Language-Action models (VLA). It uses a Franka robot. It consists of 4 different sub-benchmarks (Spatial, Object, Goal, and Long), each comprising 10 tasks. Among them, the tasks in the Libero-Long consist of 2 sequential subtasks, where the appropriate next action depends on whether the previous step has been completed or not. In addition, many tasks in Libero-Spatial, Object, Goal benchmark are simple, short-horizon pick-and-place with multi-view camera setups, which can be used to evaluate whether multi-frame observations do not degrade performance in Markovian tasks. While each sub-benchmark has its own training dataset (Team et al., 2024; Kim et al., 2024), we follow the setup of $\pi_0$ (Black et al., 2024) that combines all training datasets to train the models and reports the performance separately for each sub-benchmark. We use a fixed front-view camera and a wrist camera of 224×224 resolution without depth. We use the end-effector position as the action mode. We train our method on the combined training dataset of Libero with 3 random seeds, evaluate each model 50 times for each task by varying the position of objects, and report the average success rates with a 95% confidence interval.

**Simpler-WidowX.**  We consider Simpler-WidowX Benchmark (Li et al., 2024b), a more challenging simulated benchmark. It uses the WidowX robot. It consists of 4 tasks (Spoon on Towel, Carrot on Plate, Stack Cube, and Put Eggplant in Basket). We follow the common setup Li et al. (2024b) that uses the BridgeV2 dataset (Walke et al., 2023) to fine-tune the model, and then evaluate the models in this benchmark. We use the primary camera in the BridgeV2 dataset when training, and use a fixed third-person view camera of 224×224 resolution without depth. We use the end-effector position as the action mode. Here, because the policy uses only a third-person view, which leads to partial observability as some parts of the robot arm often move outside the camera, and objects become occluded during manipulation. We train our method on the BridgeV2 dataset with 3 random seeds, and evaluate each model 50 times for each task by varying the random seed of this benchmark, and report the average success rates with a 95% confidence interval.

**Robocasa.** We additionally consider Robocasa benchmark (Nasiriany et al., 2024), which includes 24 tasks in simulated kitchen environments. We randomly sample 100 demonstrations per task in the machine-generated training dataset in Robocasa and combine all these demonstrations to train the models. We use a fixed left-view camera, a fixed right-view camera, and a wrist camera of $224 \times 224$ resolution without depth. We use the end-effector position as the action mode. We train our method on the training dataset of Robocasa with 3 random seeds, evaluate models 50 times for each task with random seeds, and report the average success rates with a 95% confidence interval.

## B.2 REAL-WORLD ROBOTIC TASKS

We design several real-world robotic tasks. We collect 50 demonstrations per task and report the performance of the models by fine-tuning VLAs on each task. We evaluate models 20 times for each task. In the below, we describe the detailed setups (see Figure 6 for a visualization of the collected train demonstrations):

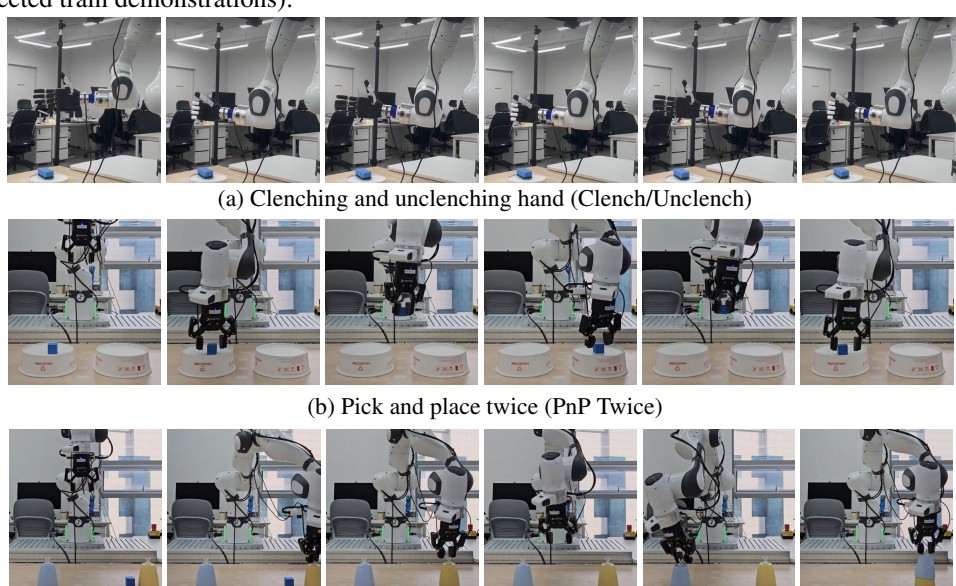

(a) Clenching and unclenching hand (Clench/Unclench)

(b) Pick and place twice (PnP Twice)

(c) Cover the cube and stack up (CoverNStack)

Figure 6: **Visualization of real-world robotic task demonstrations.**

**Clench/Unclench.** The policy should clench and unclench the right hand of a humanoid robot repeatedly. For the robot setup, we use Franka Research 3 (Robot arm) + Inspire Dex hands RH56FTP (Robot hand). We use the third-person view camera and wrist camera of $224 \times 224$ resolution without depth. We use an absolute end-effector position as the action mode. We use the text instruction "*Clench and then unclench the hand repeatedly.*" for this task. We report success if the robot clenching and unclenching a hand at least five times within 1 minute.

**PnP Twice.** Given a cube and two plates (A and B), the policy should move the cube from plate A to B and then back from B to A. We use Franka Research 3 (Robot arm) + DROID gripper. We use the third-person view camera and wrist camera of $720 \times 1280$ resolution without depth. We use the visual observation after resizing it to $224 \times 224$ resolution. We use absolute joint position as action mode. We use the text instruction "*Pick up the cube and place it on the opposite side, and then return it to the original side.*" for this task. We report partial success if the robot move the cube to the opposite side once, and full success if the robot complete the task within 1 minute.

**CoverNStack.** Given a cube and two cups, the policy should cover the cube with the closest cup, and then stack the other cup on top of the covered cup. We use the same robot, camera setup, and action mode as in the PnP Twice task. We use the text instruction "*Cover the cube with the nearest cup, then stack the other cup on top of it.*" for this task. We report partial success if the robot covers the cube, and full success if the robot complete the task within 1 minute.

## C  BASELINES

We describe the main idea of baseline methods that we used for the evaluation.

- **Octo** (Team et al., 2024) processes visual observations and task instructions to produce a read-out token, which is fed into a diffusion model to generate actions.
- **OpenVLA** (Kim et al., 2024) fine-tunes a pretrained VLM to autoregressively generate action tokens from visual observations and task instructions.
- **RoboVLMs** (Liu et al., 2025) uses multi-frame observations to generate action. It obtains tokens from each frame via the VLM and then concatenates them to generate the action.
- **TraceVLA** (Zheng et al., 2024) first extracts an image of the robot trajectories from multi-frame observation using point tracking models, and then uses this image to generate action via autoregressive modeling.
- **SpatialVLA** (Qu et al., 2025) integrates 3D spatial understanding capability into VLAs by using Ego3D position encoding and adaptive action grids.
- **NORA** (Hung et al., 2025) uses Qwen2.5-VL (Bai et al., 2025) as a backbone model to generate discrete action tokens. It decodes the token to action values using the FAST tokenizer.
- $\pi_0$ (Black et al., 2024) proposes a diffusion-based VLA that shares self-attention layers between VLM and diffusion transformer.
- $\pi_0$-**FAST** (Pertsch et al., 2025) is an autoregressive VLA utilizing Frequency-space Action Sequence Tokenization (FAST) action tokenizer. The tokenizer applies the discrete cosine transform (DCT) algorithm for encoding the continuous action values to discrete tokens. Then, the FAST tokenizer applies Byte Pair Encoding (BPE) to compress sequences.
- **GR00T N1.5** (GEAR, 2025) also proposes diffusion-based VLA, but it feeds the last hidden states of the VLM into the cross-attention layer of the diffusion transformer.

## D  RESULTS ON CALVIN BENCHMARK

To demonstrate that our method can perform long-horizon tasks, we evaluate our method by fine-tuning pre-trained VLAs on the CALVIN (ABC→D) benchmark.

**Setup.**  CALVIN (ABC→D) benchmark (Mees et al., 2022) is a simulated robotic manipulation benchmark designed to evaluate the ability of policy in a zero-shot long-horizon task. It consists of 34 distinct tasks. We train both $\pi_0$ and $\pi_0$ + ContextVLA on the training dataset collected from environments A, B, and C for 60K iterations with a batch size of 32. We then evaluate each model on the environment D with 1000 trials and report the average number of successes among 5 sequential subtasks.

**Results.**  As shown in Table 8, ContextVLA significantly improves $\pi_0$, for example, ContextVLA achieves success rates of 69% in completing all 5 tasks consecutively, whereas $\pi_0$ achieves 60%. This demonstrates the effectiveness of ContextVLA on long-horizon tasks.

Table 8: **Results on CALVIN (ABC→D).** We report the success rates of consecutive completions (%) and average number of successes among 5 sequential subtasks (Avg. success length) of VLAs fine-tuned on the training dataset of CALVIN benchmark collected from environments A, B, and C.

| Method | Success rates of consecutive completions (%) | | | | | Avg. success length |
| --- | --- | --- | --- | --- | --- | --- |
| | 1 / 5 | 2 / 5 | 3 / 5 | 4 / 5 | 5 / 5 | |
| $\pi_0$ (Black et al., 2024) | 90.95 | 82.53 | 74.00 | 66.95 | 59.58 | 3.740 |
| **+ ContextVLA (Ours)** | **93.38** | **92.70** | **85.80** | **77.58** | **69.36** | **4.238** |

# E ADDITIONAL ANALYSIS

**Amortization Strategy.** We investigate the amortization strategy for obtaining a context token in ContextVLA. To evaluate this, we fine-tune $\pi_0$ by compressing past observations at a middle block using three compression methods: global average pooling, Perceiver Resampler (Alayrac et al., 2022; Jain et al., 2024), and Attention Pooling (Ryoo et al., 2024). In Table 9, we find that when compressing past observations into a single token, average pooling performs the best. For instance, on the Simpler-WidowX benchmark, ContextVLA with average pooling achieves 56.2%, whereas using a Perceiver Resampler yields only 53.0% when compressing the observations into a single token. Notably, Perceiver Resampler needs 64 tokens to show the comparable performance of global average pooling, highlighting the efficiency of our method in extracting temporal context with extremely compressed token.

Table 9: Analysis of Amortization methods with different number of compressed tokens on Simpler-WidowX benchmark (Li et al., 2024b). All models are $\pi_0$ (Black et al., 2024) finetuned for 60K iterations with a batch size of 32, using the BridgeV2 dataset (Ebert et al., 2021). We report the averaged success rates (%) of the models on the benchmark.

| Amortization | # tokens | Simpler-WidowX |
|---|---|---|
| Perceiver Resampler | 1 | 53.0 |
| | 8 | 53.0 |
| | 64 | 56.0 |
| Attention Pooling | 1 | 52.0 |
| | 8 | 51.5 |
| | 64 | 53.0 |
| Average Pooling | 1 | **56.2** |
| | 8 | 56.0 |
| | 64 | 54.5 |
| No compression | 2048 | 47.8 |

**The Number of Amortized Context Token.** We investigate the appropriate number of context tokens for utilizing the temporal context of historical observations to generate actions. To evaluate this, we fine-tune $\pi_0$ with compressing past observations into $m = 1$, 8, or 64 tokens, by reshaping the flattened tokens of length $M$ into a tensor of shape $(m, M/m)$, and then applying average pooling along the second dimension to obtain $m$ tokens. In Table 9, we find that compressing historical observations into a single token is the optimal choice, while compressing historical observations into multiple context tokens still consistently outperforms the baseline $\pi_0$ that processes 8-frame observations without compression.

**Visualization of Similarity of Amortized Context Token.** To analyze what information is preserved in the amortized context token, we calculate the cosine similarity of the context tokens and visualize the normalized similarity matrix. Specifically, we calculate the similarity matrix of context tokens obtained during conducting tasks from the Libero benchmark. In Figure 7, we find that similar movement of the robot induces similar context tokens, indicating that context token preserves the motion context of the robot rather than the static background or object.

# F DETAILED ACTION GENERATION PROCESS

We here describe the detailed process of action generation of VLAs used in our framework: Autoregressive modeling (Kim et al., 2024; Pertsch et al., 2025), and diffusion-based modeling (Black et al., 2024; Bjorck et al., 2025).

## F.1 AUTOREGRESSIVE MODELING.

Autoregressive VLAs (O'Neill et al., 2024; Pertsch et al., 2025) encode a continuous robot action $\mathbf{a}_{t:t+l}$ into discrete action tokens using an action tokenizer (Bu et al., 2025b; Pertsch et al., 2025), and then train a Vision-Language Model (VLM) to generate the discrete action tokens via next token prediction. Concretely, We first use the visual observations $\mathbf{o}_{t-k:t}$ and the text instruction $\mathbf{c}_t$ as a prompt for the VLM. And then, VLM autoregressively generates tokens, where we consider the tokens as discretized action values. After the tokens are generated, we decode the tokens into continuous action by using action de-tokenizer.

In our experiments, we use $\pi_0$-FAST, which uses FAST tokenizer (Pertsch et al., 2025) for encoding and decoding actions. The FAST tokenizer improves the compactness and expressivity of the discretized action space by tokenizing actions in the frequency domain. Specifically, actions are

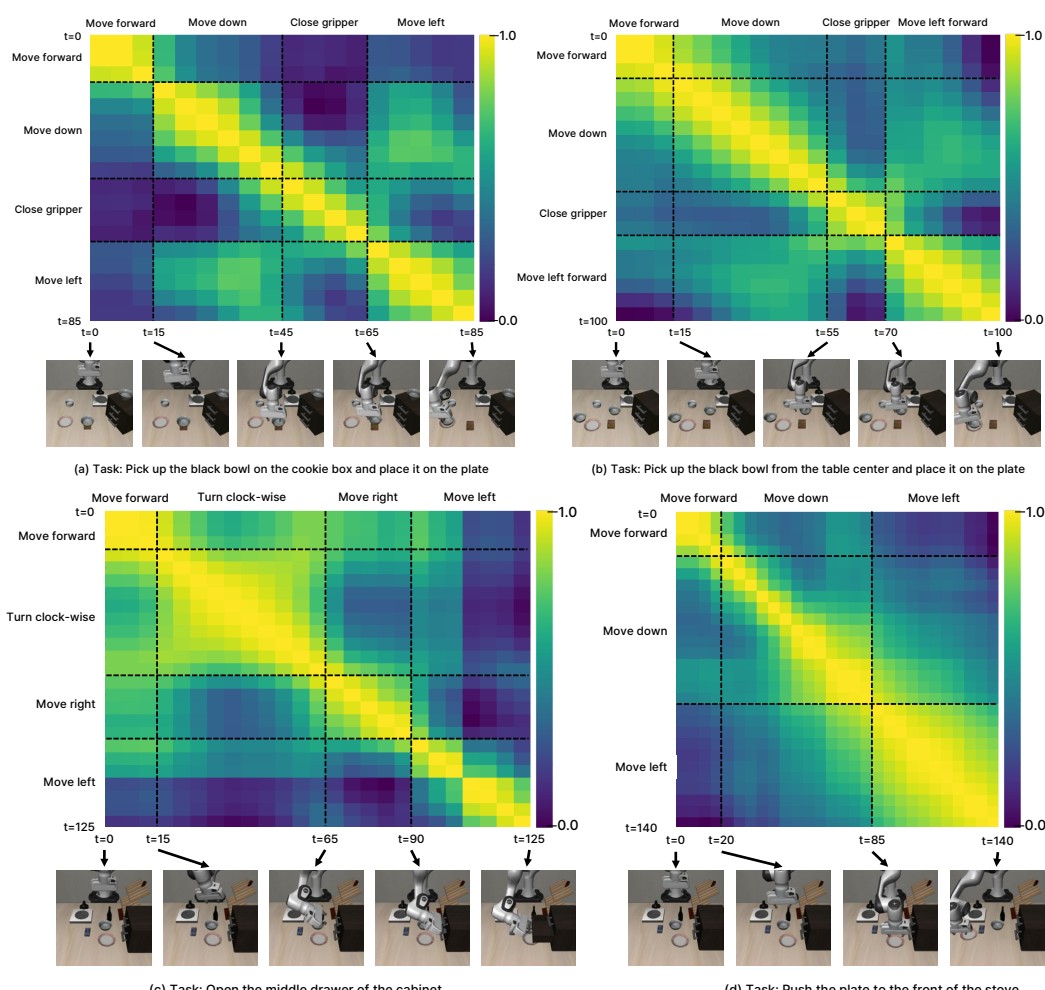

Figure 7: Visualization of the normalized cosine similarity matrix of context tokens obtained during conducting tasks from the Libero benchmark. We highlight the movement of the robot during the evaluation and the resulting visual observations on the matrix.

first transformed by a discrete cosine transform (DCT; Ahmed et al. 2006), then quantized and compressed into discrete tokens via byte-pair encoding (BPE; Gage 1994). These tokens are assigned to unused special tokens in the vocabulary of the VLM for training and generation.

### F.2 DIFFUSION-BASED MODELING.

Diffusion-based VLAs (Black et al., 2024; Bjorck et al., 2025) generates action using diffusion model conditioned on a VLM. We first process the visual observation $\mathbf{o}_{t-k:t}$ and text instruction $\mathbf{c}_t$ to extract high-level semantic information using VLM. Then, the extracted features are used as conditioning for Diffusion Transformer (DiT; Peebles & Xie 2022). Here, there are many approaches to condition the features on DiT, *e.g.*, $\pi_0$ (Black et al., 2024) shares self-attention layers between VLM and DiT, and GR00T N1.5 (Bjorck et al., 2025) feeds the last hidden states of the VLM into cross-attention layer of DiT.

During Training, we sample denoising timestep $\tau \in [0, 1]$. Then, for the target action chunk $\mathbf{a}_{t:t+l}$ of the robot (Zhao et al., 2023; Chi et al., 2023), we add noise to the action using Gaussian noise $\epsilon \sim \mathcal{N}(\mathbf{0}, \mathbf{I})$:

$$\mathbf{a}_{t:t+l}^{\tau} = \tau \mathbf{a}_{t:t+l} + (1 - \tau)\epsilon. \tag{1}$$

Then, VLA model $\pi_\theta$ processes visual observation $\mathbf{o}_t$ and text instruction $\mathbf{c}_t$ to approximate the denoising direction, $i.e.$, $\epsilon - \mathbf{a}_{t:t+l}$ by minimizing flow-matching loss:

$$\mathcal{L} = \mathbb{E}_\tau \left[ \left\| \pi_\theta(\mathbf{a}_{t:t+l}^\tau | \mathbf{o}_{t-k:t}, \mathbf{c}_t) - (\epsilon - \mathbf{a}_{t:t+l}) \right\|^2 \right]. \tag{2}$$

During inference, we generate action $\mathbf{a}_{t:t+l}$ through denoising $N$ steps. We sample random noise $\mathbf{a}_{t:t+l}^0 \sim \mathcal{N}(\mathbf{0}, \mathbf{I})$, and apply ODE or SDE sampler to denoise it to generate action $\mathbf{a}_{t:t+l}$.

## G WEIGHT-INITIALIZATION ANALYSIS

In Figure 8, we analyze how different parameter initializations affect policy performance when using multi-frame observation. We consider two architectures, $\pi_0$ (Black et al., 2024) and GR00T N1.5 (GEAR, 2025), and train them under three initialization schemes: (1) For ViT-init, only a vision encoder is initialized with a pre-trained model (Zhai et al., 2023; Tschannen et al., 2025). (2) For VLM-init, the vision encoder and VLM backbone are initialized with a pre-trained VLM (Beyer et al., 2024; Chen et al., 2025). (3) For VLA-init, entire model is initialized with a pre-trained VLA, $i.e.$, $\pi_0$ or GR00T N1.5. We observe that a policy model of VLA architecture but initialized only with a vision encoder ($i.e.$, ViT-init) suffers from performance degradation when using multi-frame inputs. In contrast, policies initialized with a pre-trained VLM or with VLA alleviate or even overcome this issue, indicating the key factor in mitigating the problem is leveraging pre-trained Vision-Language Model (VLM) to extract information for action generation.

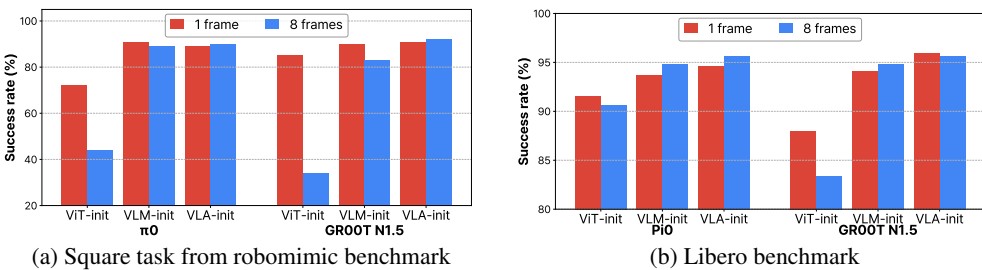

(a) Square task from robomimic benchmark                    (b) Libero benchmark

Figure 8: Success rates (%) of policy models of VLA architecture trained using either 1-frame or 8-frame observations under different weight initialization schemes. ViT, VLM, and VLA-init indicate how the VLA architecture is initialized for training; we use a pre-trained vision encoder, VLM, or VLA, respectively, and other parameters are randomly initialized.

## H ADDITIONAL REALTED WORK

**Multi-frame Observations for Behavior Cloning Policy**    Many studies have observed that learning behavior cloning (BC) policy with multi-frame observation can lead to performance degradation (Muller et al., 2005; Bansal et al., 2019; Wang et al., 2019; Codevilla et al., 2019; De Haan et al., 2019; Wen et al., 2020; Spencer et al., 2021; Seo et al., 2023; Torne et al., 2025). Early works have shown that policies trained on multi-frame observations are prone to learn correlated features that do not causally determine the expert actions ($i.e.$, spurious correlation; De Haan et al. 2019; Wen et al. 2020; Spencer et al. 2021; Seo et al. 2023). This often produces the so-called copycat behavior (Wen et al., 2020), where the policy ignores the observations and simply mimics the previous actions as its next action. However, recently, Torne et al. (2025) has observed that diffusion policy (Chi et al., 2023) shows a different trend to the traditional policies, where the model underutilizes the past action, rather than mimicking the previous actions. In contrast, our work finds that recent Vision-Language-Action (VLA) models show the opposite trend: they do not show the performance degradation commonly observed in traditional BC policies. Moreover, we introduce a method that effectively leverages multi-frame observations.

## I    QUALITATIVE RESULTS

In Figure 9, we provide qualitative results for the real-world robotic tasks. We find that a policy that uses single-frame observations often fails to determine the correct next action, leading to failure in many cases, but ContextVLA leverages multi-frame observations to perform the task successfully.

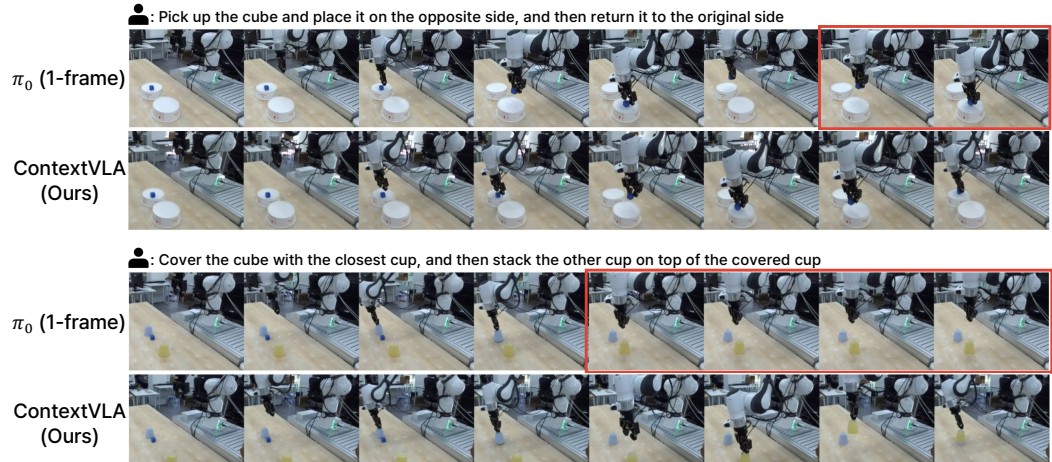

Figure 9: **Qualitative results.** $\pi_0$ that uses single-frame observations fails to determine the correct next action due to the lack of utilizing temporal context (in red box), but ContextVLA leverages temporal context to determine the correct next action depending on the previous movement of a robot.

## J    ADDITIONAL QUANTITATIVE RESULTS

In Tables 10 and 11, we report performance of our method compared to the VLAs that uses 8-frame visual observations. We find that ContextVLA achieves performance comparable to, or even surpassing, the VLAs that use 8-frame observation without compression. This indicates that ContextVLA effectively compresses past observations into an amortized token that captures the temporal context well.

## K    USE OF AI TOOLS

We acknowledge that a large language model (LLM) was used to refine the phrasing and grammar of the manuscript.

Table 10: **Results on Libero.** We report the success rates (%) of various VLAs fine-tuned on the training dataset of Libero (Liu et al., 2023a).

| Method | # frames | Spatial | Object | Goal | Long | Avg. |
|---|---|---|---|---|---|---|
| $\pi_0$ (Black et al., 2024) | 1 | 96.3 | 97.3 | 96.2 | 88.8 | 94.7 |
| $\pi_0$ (Black et al., 2024) | 8 | 97.2 | 98.4 | 94.6 | 92.0 | 95.6 |
| **+ ContextVLA (Ours)** | 8 | **97.9** | **98.9** | **96.3** | **93.1** | **96.6** |
| $\pi_0$-FAST (Pertsch et al., 2025) | 1 | 96.3 | 97.5 | 94.5 | 84.8 | 93.3 |
| $\pi_0$-FAST (Pertsch et al., 2025) | 8 | **98.0** | **99.2** | **96.8** | **91.2** | **96.3** |
| **+ ContextVLA (Ours)** | 8 | 97.8 | 98.9 | 95.9 | 90.8 | 95.8 |
| GR00T N1.5 (GEAR, 2025) | 1 | 98.0 | 99.3 | 96.9 | 88.7 | 95.7 |
| GR00T N1.5 (GEAR, 2025) | 8 | 98.4 | **99.4** | 95.8 | 88.6 | 95.6 |
| **+ ContextVLA (Ours)** | 8 | **98.6** | 99.1 | **97.3** | **93.0** | **97.0** |

Table 11: **Results on Simpler-WidowX.** We report the success rates (%) of various VLAs fine-tuned on the Bridgev2 dataset (Walke et al., 2023).

| Method | # frames | Spoon on Towel | Carrot on Plate | Stack Cube | Put Eggplant in Basket | Avg. |
|---|---|---|---|---|---|---|
| $\pi_0$ (Black et al., 2024) | 1 | 46.5 | 38.7 | 42.7 | 39.3 | 41.8 |
| $\pi_0$ (Black et al., 2024) | 8 | 41.3 | 42.7 | **43.3** | 64.0 | 47.8 |
| **+ ContextVLA (Ours)** | 8 | **53.3** | **56.0** | 41.3 | **74.0** | **56.2** |
| $\pi_0$-FAST (Pertsch et al., 2025) | 1 | 59.0 | 79.0 | 65.0 | 33.0 | 59.0 |
| $\pi_0$-FAST (Pertsch et al., 2025) | 8 | 58.0 | 58.0 | **90.0** | 52.0 | 64.5 |
| **+ ContextVLA (Ours)** | 8 | **60.7** | **81.3** | 78.7 | **62.0** | **70.7** |
| GR00T N1.5 (GEAR, 2025) | 1 | **30.0** | 28.0 | **16.0** | 42.7 | 29.2 |
| GR00T N1.5 (GEAR, 2025) | 8 | 8.0 | 2.0 | 2.0 | 8.0 | 5.0 |
| **+ ContextVLA (Ours)** | 8 | 28.0 | **29.3** | 14.7 | **50.3** | **31.8** |

Table 12: **Full Results on Robocasa.** We report the success rates (%) of various VLAs fine-tuned on the training dataset of Robocasa (Nasiriany et al., 2024), consisting of 24 tasks with 100 demos per task

| Method | # frames | Pick and Place | Others | Avg. |
|---|---|---|---|---|
| $\pi_0$ (Black et al., 2024) | 1 | 32.9 | 70.3 | 57.9 |
| $\pi_0$ (Black et al., 2024) | 8 | **35.8** | **70.8** | **59.1** |
| **+ContextVLA (Ours)** | 8 | 35.6 | 70.4 | 58.8 |
| $\pi_0$-FAST (Pertsch et al., 2025) | 1 | 46.1 | 68.7 | 62.0 |
| $\pi_0$-FAST (Pertsch et al., 2025) | 8 | 40.1 | **69.7** | 59.8 |
| **+ContextVLA (Ours)** | 8 | **48.6** | 68.7 | **62.0** |
| GR00T N1.5 (GEAR, 2025) | 1 | 51.8 | 67.6 | 62.3 |
| GR00T N1.5 (GEAR, 2025) | 8 | **53.0** | 67.0 | 62.3 |
| **+ ContextVLA (Ours)** | 8 | 52.8 | **70.2** | **64.4** |

