# OpenReview forum: "ContextVLA: Vision-Language-Action Model with Amortized Multi-Frame Context"
_ICLR.cc/2026/Conference — Submitted to ICLR 2026_

### Official Review · Reviewer_2Jbw · 2025-10-28

**Soundness:** 2
**Presentation:** 2
**Contribution:** 2
**Rating:** 2
**Confidence:** 4

**Summary:**

This work enables the usage of multi-frame visual observations for action generation in VLAs, while mitigating common problems that come with history-conditioned policies such as causal confusion. The proposed approach is additionally computationally efficient.

**Strengths:**

The paper attempts to address an important problem that has been studied extensively. The proposed approach seems simple yet effective, and is applied to a good variety of open-source VLAs as well as benchmarks. The empirical performances seem decent, and the ablation studies are done well.

**Weaknesses:**

This paper addresses the problem of leveraging multi-frame visual observation history for making model predictions. This is a widely studied problem under many names, such as causal confusion, copycat problem, shortcut learning, etc. There needs to be better citations of this line of work, such as the early works of [1] that and the more recent instantiation in [2].

While the reported mean success rates seem impressive, the lack of any statistical significance or uncertainty quantification severely weakens the results. I would recommend the authors to add standard deviations / errors to the figures and the tables.

Finally, while it's shown that the method improves policy performance in non-Markovian tasks across the 3 benchmarks.. There should be 1) an explainer on the tasks themselves and how they exhibit non-Markovian properties and 2) that there should be some evidence that the method does not hurt performance in Markovian tasks.

I would be willing to raise my score if the concerns here are addressed.

[1] https://papers.nips.cc/paper_files/paper/2005/hash/fdf1bc5669e8ff5ba45d02fded729feb-Abstract.html
[2] https://ieeexplore.ieee.org/document/9009463

**Questions:**

See section on weaknesses.

---

> ### Author Response · Authors · 2025-11-21
> **Response to Reviewer 2Jbw (1/2)**
>
> Dear Reviewer 2Jbw,
>
> We sincerely appreciate your insightful comments and efforts in reviewing our manuscript. We address your comments one-by-one below. In the revised draft, we mark our major revisions as “blue”.
>
> ---
>
> **[W1] Lack of citations: Problem of leveraging multi-frame observations for policies.**
>
> **[A1]** Thank you for the suggestion. While the problem of leveraging multi-frame observation is well-known for traditional behavior cloning policies, these discussions have not been extended to recent Vision-Language-Action (VLA) models. An interesting finding in this paper is that recent VLAs, especially ContextVLA, show a different trend. As shown in Figure 2, VLAs already benefit from multi-frame observations compared to single-frame observations. Furthermore, we show that ContextVLA consistently outperforms 1-frame baselines on diverse benchmarks in an architecture-agnostic manner, demonstrating that ContextVLA effectively captures temporal information rather than degrading performance due to learning shortcut or copycat behavior. Nevertheless, we add more related works regarding the problem you mentioned [1-4] in the revision (see Appendix H).
>
> [1] Muller et al., Off-Road Obstacle Avoidance through End-to-End Learning, NeurIPS 2005 \
> [2] Bansal et al., ChauffeurNet: Learning to Drive by Imitating the Best and Synthesizing the Worst, RSS 2019 \
> [3] Wang et al., Monocular Plan View Networks for Autonomous Driving, IROS 2019 \
> [4] Codevilla et al., Exploring the Limitations of Behavior Cloning for Autonomous Driving, ICCV 2019
>
> ---
>
> **[W2] Statistical significance or uncertainty quantification.**
>
> **[A2]** Following your helpful suggestion, we additionally train each model with 3 different random seeds and report the mean success rate with a standard deviation. As shown in the tables below, the improvement is still statistically significant, e.g., on the Libero-Long benchmark, $\pi_0$-FAST gets success rates (%) of 84.8$\pm$1.4, while ContextVLA gets 90.8%$\pm$0.9. We also updated Tables 1, 2, and 3 accordingly in the revision. Moreover, we add a standard deviation to Figure 1 (b) in the revision.
>
> \begin{array}{lccccc|ccccc|ccc}
> \hline
> & & & \text{Libero} & & & & & \text{Simpler-WidowX} & & & & \text{Robocasa} \newline
> \hline
> \text{Method} & \text{Spatial} & \text{Object} & \text{Goal} & \text{Long} & \text{Avg} & \text{Spoon} & \text{Carrot} & \text{Stack} & \text{Eggplant} & \text{Avg} & \text{Pick\\&Place} & \text{Others} & \text{Avg} \newline
> \hline
> \pi_0 & 96.3\pm0.3 & 97.3\pm0.4 & 96.2\pm0.3 & 88.8\pm0.3 & 94.7\pm0.1 & 46.7\pm3.3 & 38.7\pm7.1 & 42.7\pm3.3 & 39.3\pm8.4 & 41.8\pm3.2 & 32.9\pm0.4 & 70.3\pm1.4 & 57.9\pm0.9 \newline
> \text{+ContextVLA} & \bf{97.9}\pm0.5 & \bf{98.9}\pm0.6 & \bf{96.3}\pm0.3 & \bf{93.1}\pm0.6 & \bf{96.6}\pm0.1 & \bf{53.3}\pm1.8 & \bf{56.0}\pm2.3 & \bf{41.3}\pm2.7 & \bf{74.0}\pm7.2 & \bf{56.2}\pm1.8 & \bf{35.6}\pm0.8 & \bf{70.4}\pm0.2 & \bf{58.8}\pm0.2 \newline
> \hline
> \pi_0\text{-FAST} & 96.3\pm1.1 & 97.5\pm0.9 & 94.5\pm0.6 & 84.8\pm1.4 & 93.3\pm0.5 & 59.0\pm1.2 & 79.0\pm1.2 & 65.0\pm1.9 & 33.0\pm1.0 & 59.0\pm0.7 & 46.1\pm1.0 & 68.1\pm0.8 & 60.3\pm0.1 \newline
> \text{+ContextVLA} & \bf{97.8}\pm0.6 & \bf{98.9}\pm0.4 & \bf{95.9}\pm0.6 & \bf{90.8}\pm0.9 & \bf{95.8}\pm0.2 & \bf{60.7}\pm1.5 & \bf{81.3}\pm5.4 & \bf{78.7}\pm4.3 & \bf{62.0}\pm9.2 & \bf{70.7}\pm1.7 & \bf{48.6}\pm0.6 & \bf{68.7}\pm0.6 & \bf{62.0}\pm0.2 \newline
> \hline
> \text{GR00T N1.5} & 98.0\pm0.5 & \bf{99.3}\pm0.2 & 96.9\pm0.3 & 88.7\pm0.3 & 95.7\pm0.2 & \bf{30.0}\pm1.5 & 28.0\pm1.2 & \bf{16.0}\pm3.5 & 42.7\pm1.8 & 29.2\pm0.9 & 51.8\pm1.4 & 67.6\pm1.1 & 62.3\pm0.3 &  \newline
> \text{+ContextVLA} & \bf{98.6}\pm0.2 & 99.1\pm0.2 & \bf{97.3}\pm0.1 & \bf{93.0}\pm0.3 & \bf{97.0}\pm0.1 & 28.0\pm2.3 & \bf{29.3}\pm0.7 & 14.7\pm3.3 & \bf{50.3}\pm2.4 & \bf{31.8}\pm1.5 & \bf{52.8}\pm0.5 & \bf{70.2}\pm0.5 & \bf{64.4}\pm0.2 \newline
> \hline
> \end{array}

---

> ### Author Response · Authors · 2025-11-21
> **Response to Reviewer 2Jbw (2/2)**
>
> **[W3-1] There should be an explanation of how the tasks exhibit non-Markovian properties.**
>
> **[A3-1]** We first clarify that the non-Markovian properties of our real-world tasks are already explained in Section 3.3.  Each task requires the policy to decide actions based on previous movements (e.g., deciding whether to move left or right during repeated pick-and-place), which cannot be inferred from a single frame.
> Regarding the simulated robotic manipulation benchmarks, they also exhibit such characteristics in several tasks. For instance, in the Simpler-WidowX benchmark, the policy observes only a third-person view, which leads to partial observability because some parts of the robot arm often move outside the camera, and objects become occluded during manipulation. In addition, the tasks in the Libero-Long benchmark consist of two sequential sub-tasks, where the appropriate next action depends on whether the previous sub-task has been completed or not. This is not always visually apparent in the single frame, so the policy must rely on the history to determine the correct action. We include these explanations in the revision (see Appendix B.1).
>
> ---
>
> **[W3-2] There should be some evidence that ContextVLA does not hurt performance in Markovian tasks.**
>
> **[A3-2]** Thank you for pointing this out. However, we clarify that several tasks in the simulated robotic manipulation benchmarks are inherently Markovian (not non-Markovian), and our results already demonstrate that ContextVLA does not degrade performance in these settings. For example, in Libero-Spatial, -Object, and -Goal benchmarks, many tasks are simple, short-horizon pick-and-place with multi-view camera setups, which generally makes the task Markovian. ContextVLA achieves near-perfect success rates on these tasks and shows no performance degradation compared to the baselines. We highlight these results and include this discussion in the revision (see Appendix B.1).

---

> ### Author Response · Authors · 2025-11-28
> **Gentle Reminder**
>
> Dear Reviewer 2Jbw,
>
> Thank you again for your time and thoughtful efforts in reviewing our paper.
>
> As the discussion period comes within a week of its end, we would like to gently remind you in case you have any remaining comments. We believe that we have sincerely and successfully addressed your concerns, supported by the corresponding additional experimental results.
>
> If you have any further concerns or questions, please feel free to let us know.
>
> Thank you very much \
> Authors

---

### Official Review · Reviewer_XBWu · 2025-10-28

**Soundness:** 2
**Presentation:** 3
**Contribution:** 2
**Rating:** 4
**Confidence:** 2

**Summary:**

This paper introduces ContextVLA, a Vision-Language-Action model that effectively leverages multi-frame observations to improve robotic task performance.
The core contribution of this paper is a mechanism to compress past observations into a single context token at first n blocks in the VLM. For the remaining N-n blocks, the current observation just attends the single context token instead of the long raw observation tokens to realize efficient training and inference.
The authors conduct experiments on diverse benchmarks to show the effectiveness of the proposed method.

**Strengths:**

1. The method is simple and easy to follow.
2. The writing is clear.
3. SOTA performance on diverse benchmarks.

**Weaknesses:**

I am not an expert in this field, so I tend to see other reviewers' comments as well as the rebuttal to make my final decisions. Below are the weaknesses of this paper from my perspective:

1. **The effectiveness of ContextVLA is unclear.**
Since video LLMs generally require a huge amount of pre-training data to achieve training convergence **under long context scenarios**, e.g., LLaVA-video and Qwen2.5-VL, fine-tuning single-frame-based $\pi_0$ on extremely small-scale downstream tasks is hard to reach convergence. Instead, overfitting is very likely to happen, which may be the reason $\pi_0$ with 8 frames lags behind ContextVLA.
And this can be evidenced by the counterintuitive results in Table 4, where $\pi_0$ finetuned with 8 frames has nearly no gains.
Since ContextVLA is claimed as a generalized approach, applying it to other multi-frame based methods, e.g., Octo and RoboVLMs, should make this method more pronounced.

2. **Lack of robustness evaluation.**
The authors should report the performance where the inference frames differ from the training setting to verify if ContextVLA truly learned to summarize the context, or just overfit to the 8 frames-to-1 token paradigm. For instance, what would happen if we use 2 frames or 32 frames during inference?

3. **Lack of sufficient ablation studies.**
The results show a consistent improvement from 1 frame to 8 frames. I wonder if this rule holds for longer frames? For instance, does ContextVLA work for 32, 64, or 110 frames? If not, does it mean leveraging multi-frames is not always beneficial?

**Questions:**

Please refer to "Weakness" for details.

---

> ### Author Response · Authors · 2025-11-21
> **Response to Reviewer XBWu**
>
> Dear Reviewer XBWu,
>
> We sincerely appreciate your insightful comments and efforts in reviewing our manuscript. We address your comments one-by-one below. In the revised draft, we mark our major revisions as “blue”.
>
> ---
>
> **[W1] Effectiveness of ContextVLA compared to a naive 8-frame baseline is unclear.**
>
> **[A1]** While directly finetuning $\pi_0$ and GR00T N1.5 with 8-frame inputs may not fully exploit temporal information, as the reviewer pointed out, we would like to clarify that it does not indicate the unclear effectiveness of ContextVLA. First, these VLAs already utilize video pre-trained VLMs (e.g., $\pi_0$ uses PaliGemma), thus they should understand videos to generate actions. Second, the effectiveness of ContextVLA compared to a naive 8-frame baseline is clear in terms of inference latency, which is an important problem in robotic tasks because slow inference often degrades the performance of real-robotic manipulation tasks [1].
>
> For applying ContextVLA to multi-frame-based methods, we first note that the methods that you mentioned are orthogonal to our method, because (1) the VLM backbone of RoboVLMs processes only a single frame and later concatenates per-frame outputs to generate actions while our method compresses multi-frame observations directly inside the VLM backbone, and (2) Octo processes only 2-frames, thus compressed context token of historical observation does not include temporal context.
>
> Instead, we compare video-based VLAs pretrained on the OXE dataset by pretraining ContextVLA on the OXE dataset. We evaluate the models by finetuning them on downstream manipulation tasks. As shown in the table below, we find that ContextVLA outperforms the video-based VLAs, demonstrating the effectiveness of ContextVLA over other multi-frame-based methods. We include the results in Tables 1 and 2 in the revision.
>
> \begin{array}{lcccccc}
> \hline
> &&&&\text{Libero} \newline
> \hline
> \text{Method} & \text{\\#frames} & \text{Spatial} & \text{Object} & \text{Goal} & \text{Long} & \text{Avg} \newline
> \hline
> \text{Octo}       & 2 & 78.9 & 85.7 & 84.6 & 51.1 & 75.1 \newline
> \text{TraceVLA}   & 6 & 84.9 & 85.2 & 75.1 & 54.1 & 74.8 \newline
> \text{ContextVLA (Ours)} & 8 & \bf{95.8} & \bf{99.2} & \bf{92.6} & \bf{87.0} & \bf{93.7} \newline
> \hline
> \end{array}
>
> \begin{array}{lcccccc}
> \hline
> &&&&\text{Simpler-WidowX} \newline
> \hline
> \text{Method} & \text{\\#frames} & \text{Spoon} & \text{Carrot} & \text{Stack} & \text{Eggplant} & \text{Avg} \newline
> \hline
> \text{Octo-base}  & 2  & 12.5 &  8.3 &  0.0 & 43.1 & 16.0 \newline
> \text{Octo-small} & 2  & 47.2 &  9.7 &  4.2 & 56.9 & 29.5 \newline
> \text{RoboVLMs}   & 16 & 29.2 & 25.0 & 12.5 & 58.3 & 31.3 \newline
> \text{ContextVLA (Ours)} & 8  & \bf{52.0} & \bf{56.0} & \bf{58.0} & \bf{72.0} & \bf{59.5} \newline
> \hline
> \end{array}
>
> ---
>
> **[W2] Robustness evaluation: Using a different number of frames at inference to verify if ContextVLA summarizes temporal context or overfits to the 8-frame setup.**
>
> **[A2]** We clarify that the robustness evaluation of the number of frames at inference time is not required. This is because the number of frames is a design choice determined during training rather than during inference, and the standard inference setup is to use the same number of frames as in training [1,2]. When fewer frames are available at inference, e.g., in very early timesteps, common practice is to repeat or pad frames to match the training setup, and we follow this by repeating the oldest frames among the input frames to get 8-frame visual observations.
>
> Moreover, the benefit of context summarization in ContextVLA does not overfit to an 8-frame setup. While ContextVLA shows the best performance on 8-frame scenarios, ContextVLA trained with a different number of frames still consistently outperforms the 1-frame baseline (see Table 5). This demonstrates that the summarization of context in ContextVLA is not merely overfit to 8-frame scenarios but can be extended to a diverse number of frames. We include this discussion in the revision (see Section 3.4, Number of Past Observations).
>
> [1] Octo Model Team, Octo: An Open-Source Generalist Robot Policy, RSS 2024 \
> [2] Li et al., Towards Generalist Robot Policies: What Matters in Building Vision-Language-Action Models, Arxiv preprint 2024
>
> ---
>
> **[W3] Lack of ablation studies: Longer frames.**
>
> **[A3]** Thank you for the comment. We agree that experiments on much longer frames are an interesting direction. However, we did not conduct the experiments because current state-of-the-art VLA backbones support only a limited maximum token length (e.g., $\pi_0$ process at most 8k tokens), and longer frames proportionally increase the computational cost at inference, making the policy impractical for real-world deployment. Nevertheless, we indeed expect that longer frames further enhance the performance, as they provide richer temporal context (e.g., the motion trajectory of a robot), which can enable the VLA to generate actions better.

---

> > ### Comment · Reviewer_XBWu · 2025-11-25
> > **Response to the rebuttal**
> >
> > Thanks for the rebuttal. But I believe most of my concerns are not raised properly. I only raise three questions, while two of them are not answered:
> >
> > **W2: Robustness evaluation**. I do not understand why the authors think this is "not required" and refer to other works. I believe this is an **inherent issue** without relating to any external models. As also pointed out by the other two reviewers (Reviewer#2Jbw: the lack of any statistical significance or uncertainty quantification severely weakens the results; Reviewer#5SdM, the paper lacks deeper analysis of what temporal or semantic information is actually preserved after pooling). **Overfitting to an 8-frame-to-1-token paradigm is of minimal practical value in real usage, especially considering that current VLMs mostly show robustness in terms of varied frame length.** Also, I am puzzled why the authors are not willing to present this experiment that only requires several re-inferences, even without training. **I can only infer that my assumption is right, the VLM is far from convergence, and it can only overfit to the 8-frame-to-1-token paradigm.**
> >
> > **W3: Lack of ablation studies: Longer frames.** As also pointed out by Reviewer#5SdM, there are many alternative VLMs that can better handle frames. **Merely presenting a monolithic-increased rule under only a limited frame setting (1,2,4,8) is not insightful for the academic community.** The authors can simply average 8 frames to 1 token and use 2 tokens to represent 16 frames, but I do not see any attempts in the rebuttal.
> >
> > **Based on the above reasons, 2/3 of my concerns are ignored by the author, and considering that 2/3 other reviewers also hold a negative attitude towards this paper (4 and 2), I decide to maintain my initial negative rating (4).**

---

> ### Author Response · Authors · 2025-11-27
> **Official Comment by Authors**
>
> We sincerely appreciate the reviewer for taking the time to provide further comments. We further address your concern one by one.
>
> ---
>
> **Robustness evaluation**
>
> For the Vision-Language Model (VLM), we fully agree with the reviewer that robustness to videos of diverse lengths is important, as users may provide arbitrary-length videos and expect the VLM to understand them correctly and respond to the question.
>
> Unlike VLMs, in robotic control with VLAs, we would like to note that the number of input frames is not an inference-time variable determined by the user, but a fixed design choice at training. During inference, visual observations arrive sequentially at a fixed rate, and the policy always receives the most recent N frames (e.g., 8 frames) rather than receiving a different number of frames. Therefore, varying the number of frames at inference time is not a typical use case, and we believe robustness evaluation regarding different numbers of frames is not required.
>
> A more important thing in practice is whether the chosen fixed-length input achieves better accuracy and faster inference, e.g., we take 8 frames based on both the success rates and inference efficiency (see [2Jbw A2] for statistical significance).
>
> Nevertheless, to address your concern, we evaluate our model ($\pi_0$ + ContextVLA) under a 2 and 4-frame inference setup by uniformly subsampling the input frames. As shown in the table below, we find that ContextVLA trained with 8-frame videos achieves performance that is comparable to, and slightly higher than, our models trained on 2-frame or 4-frame videos when evaluated with the corresponding number of frames. These results indicate that ContextVLA does not overfit to the 8-frame-to-1-token setup, and that the temporal context learned from 8-frame training provides effective information even when fewer frames are provided at inference time.
>
> \begin{array}{ccc}
> \hline
> \text{Training \\# frames} & \text{Inference \\#frames} & \text{Libero} \newline
> \hline
> 1 & 1 & 94.6 \newline
> \hline
> 2 & 2 & 94.8 \newline
> 8 & 2 & 95.0 \newline
> \hline
> 4 & 4 & 95.0 \newline
> 8 & 4 & 95.9 \newline
> \hline
> 8 & 8 & 96.6 \newline
> \hline
> \end{array}
>
> ---
>
> **Longer frames**
>
> We indeed expect that longer frames should enhance the performance, as VLAs can exploit the additional temporal context provided by the amortized context tokens derived from longer videos. For example, when 16-frame observations are compressed into 2 tokens as the reviewer mentioned, the model can utilize the additional temporal information with the context captured from 8-frame observation. This allows VLAs to benefit from a more expressive temporal context.
>
> Nevertheless, to address your concern, we additionally train $\pi_0$ + ContextVLA using 8 and 16-frame videos on the training dataset of the Libero benchmark, and evaluate the model on the Libero benchmark. For this experiment, note that we restrict the input to a single-view observation, because the default 2-view observations with 16 frames already exceed the token limits of $\pi_0$: 8192 tokens = 256 (token/view) $\times$ 2 (view/frame) $\times$ 16 (Frame). As shown in the table below, we find that increasing the number of frames from 8 to 16 improves the performance on the Libero benchmark (91.4% $\rightarrow$ 91.9%), and compressing 16 frames into 2 tokens shows slightly better performance. Notably, 16 frames especially performs better on Libero-Long (81.6% $\rightarrow$ 83.0%), which has relatively longer-horizon tasks, indicating that compressing longer temporal information into context token further benefits long-horizon tasks. This aligns with our findings in [dbTk A2], where ContextVLA shows even larger improvements on much longer-horizon scenarios.
>
> \begin{array}{ccc}
> \hline
> \text{Training \\# frames} & \text{\\# context token} & \text{Libero (view=1)} \newline
> \hline
> 8 & 1 & 91.4 \newline
> 16 & 1 & 91.9 \newline
> 16 & 2 & 92.1 \newline
> \hline
> \end{array}
>
>
> ---
>
> If you have any further questions/concerns, please do not hesitate to let us know.
>
> Thank you very much,\
> Authors

---

> > ### Comment · Reviewer_XBWu · 2025-11-28
> > **Response to the second rebuttal**
> >
> > Thanks for the second rebuttal. Now, most of my concerns are raised, and some other minor concerns are mentioned by other reviewers. At present, I treat this as a borderline paper of score 5, which means I will not vote for a clear acceptance, but if other reviewers reach a consensus that the paper should be accepted, I will not have a negative attitude towards this result.

---

### Official Review · Reviewer_5SdM · 2025-11-01

**Soundness:** 3
**Presentation:** 2
**Contribution:** 2
**Rating:** 4
**Confidence:** 2

**Summary:**

This paper proposes ContextVLA, a Vision-Language-Action (VLA) model that aims to efficiently incorporate temporal context from multiple video frames without incurring heavy computational cost. Instead of directly feeding multi-frame sequences into the model, ContextVLA adopts a two-stage amortized design: it first compresses a short history of past observations into a single context token by pooling hidden states from the VLM backbone, and then uses this token to condition downstream policy prediction. This design achieves a balance between temporal understanding and computational efficiency, allowing multi-frame reasoning at almost the same cost as single-frame inference. Experiments on multiple simulation and real-world robot benchmarks show the effectiveness of ContextVLA.

**Strengths:**

The paper targets an important problem of  VLAs — how to efficiently and effectively model temporal context. The proposed amortized multi-frame context mechanism is simple yet effective. It integrates seamlessly with existing architectures (π0, GR00T-N1, Pi0-FAST, etc.) and substantially reduces both memory footprint and inference time.

**Weaknesses:**

The paper try to tackle an important question — how to introduce history understanding into VLA models in an efficient and low-cost manner. However, the core contribution mainly lies in introducing a two-stage compression strategy that aggregates multi-frame context into a single token. While effective, this design space is rather limited and could be approached in many other ways. The proposed method is essentially a form of context compression, and alternative formulations (e.g., state-space models such as Mamba, video-representation encoders, temporal adapters, or attention pooling) could also be applied. The paper does not discuss why pooling hidden states directly from the VLM backbone is particularly advantageous over these alternatives.

Additionally, the benefit of “context tokenization” is primarily demonstrated empirically; the paper lacks deeper analysis of what temporal or semantic information is actually preserved after pooling. On several tasks, the performance gap to the base model is relatively small—might within the standard deviation range. This weakens the claim that the approach yields consistently better temporal reasoning.

**Questions:**

Please refer to the weakness part.

1. A few visualization-based analyses (e.g., token similarity, entropy, temporal variance, or per-layer contribution) could provide much stronger insight into why it works.
2. The work could also be strengthened by including ablations on Number of frames, Pooling method and number of history tokens

---

> ### Author Response · Authors · 2025-11-21
> **Response to Reviewer 5SdM (1/3)**
>
> Dear Reviewer 5SdM,
>
> We sincerely appreciate your insightful comments and efforts in reviewing our manuscript. We address your comments and questions one-by-one below. In the revised draft, we mark our major revisions as “blue”.
>
> ---
>
> **[W1] There should be a discussion on why directly pooling VLM hidden states is more advantageous than using other temporal compression methods.**
>
> **[A1]** Thank you for pointing this out. While more advanced compression modules that you mentioned may be used, we did not consider discussing them because our primary focus is more on investigating whether VLAs can leverage multi-frame temporal context through VLM representation themselves, rather than their technical details. Therefore, we use a global pooling, a well-known approach for extracting a single global context from representations [1,2] without additional parameters, and have focused on conducting experiments to demonstrate consistent improvements across diverse VLA architectures. However, we fully agree that providing such comparison could provide great insight to readers.
>
> Following your suggestions (and the Reviewer dbTk), we additionally evaluate Perceiver Resampler [1,2] and Attention Pooling [3] within our framework ($\pi_0$ + ContextVLA). As shown in the table below, we observe that when compressing past observations into a single token, global average pooling performs best (56.2% vs. 53.0% for Perceiver Resampler on Simpler-WidowX). In particular, Perceiver Resampler needs 64 tokens to achieve comparable performance to our single-token global pooling, highlighting the efficiency of our method in extracting temporal context with extremely compressed token. We include the results and discussion in the revision (see Appendix E).
>
> \begin{array}{lcc}
> \hline
> \text{Pooling method} & \\# \text{token} & \text{Simpler-WidowX (\\%)} \newline
> \hline
> \text{Perceiver Resampler} [3,4] & 1 & 53.0 \newline
>                                  & 8 & 53.0 \newline
>                                  & 64 & 56.0 \newline
> \hline
> \text{Attention Pooling} [5]     & 1 & 52.0 \newline
>                                  & 8 & 51.5 \newline
>                                  & 64 & 53.0 \newline
> \hline
> \text{\bf{Global Avg. Pooling (Ours)}} & 1 & \bf{56.2} \newline
> \hline
> \end{array}
>
> [1] Feichtenhofer et al., Masked Autoencoders As Spatiotemporal Learners, NeurIPS 2022 \
> [2] Wang et al., VideoMAE V2: Scaling Video Masked Autoencoders with Dual Masking, CVPR 2023 \
> [3] Alayrac et al., Flamingo: A Visual Language Model for Few-Shot Learning, NeurIPS 2022 \
> [4] Jain et al., Vid2Robot: End-to-End Video-Conditioned Policy Learning with Cross-Attention Transformers, RSS 2024 \
> [5] Ryoo et al., xGen-MM-Vid (BLIP-3-Video): You Only Need 32 Tokens to Represent a Video Even in VLMs, NeurIPS 2024
>
> ---
>
> **[W2/Q4] There should be a deeper analysis of what temporal or semantic information is actually preserved after pooling / visualization-based analysis**
>
> **[A2]** Following your suggestion, we analyze what information is preserved after pooling. In particular, we calculate the cosine similarity of the context tokens and visualize the normalized similarity matrix to investigate when the context tokens become similar. We include this analysis in Appendix E in the revision and include more results in the final draft.
>
> Specifically, we visualize a similarity matrix of context tokens obtained during conducting tasks from the Libero benchmark. As shown in Figure 7 in the revision, we find that similar movement of the robot induces similar context tokens, indicating that context token preserves the motion context of the robot rather than the static background or object.

---

> > ### Author Response · Authors · 2025-11-28
> > **Gentle Reminder**
> >
> > Dear Reviewer 5SdM,
> >
> > Thank you again for your time and thoughtful efforts in reviewing our paper.
> >
> > As the discussion period comes within a week of its end, we would like to gently remind you in case you have any remaining comments. We believe that we have sincerely and successfully addressed your concerns, supported by the corresponding additional experimental results.
> >
> > If you have any further concerns or questions, please feel free to let us know.
> >
> > Thank you very much \
> > Authors

---

> ### Author Response · Authors · 2025-11-21
> **Response to Reviewer 5SdM (2/3)**
>
> **[W3] The performance gap to the base model is relatively small, might within the standard deviation range.**
>
> **[A3]** We politely disagree. The performance improvement from ContextVLA is significant in several benchmarks (e.g., on the Simpler-WidowX benchmark, ContextVLA achieves 14.4% performance gain compared to $\pi_0$), while ContextVLA shows consistent improvement across all benchmarks. Nonetheless, to further address your concern, we additionally train each model with 3 different random seeds and report the mean success rate with a standard deviation. As shown in the table below, the improvement is consistent and significant, e.g., on the Libero-Long benchmark, $\pi_0$-FAST gets success rates (%) of 84.8$\pm$1.4, while ContextVLA gets 90.8%$\pm$0.9. We also updated Tables 1, 2, and 3 accordingly in the revision.
>
> \begin{array}{lccccc|ccccc|ccc}
> \hline
> & & & \text{Libero} & & & & & \text{Simpler-WidowX} & & & & \text{Robocasa} \newline
> \hline
> \text{Method} & \text{Spatial} & \text{Object} & \text{Goal} & \text{Long} & \text{Avg} & \text{Spoon} & \text{Carrot} & \text{Stack} & \text{Eggplant} & \text{Avg} & \text{Pick\\&Place} & \text{Others} & \text{Avg} \newline
> \hline
> \pi_0 & 96.3\pm0.3 & 97.3\pm0.4 & 96.2\pm0.3 & 88.8\pm0.3 & 94.7\pm0.1 & 46.7\pm3.3 & 38.7\pm7.1 & 42.7\pm3.3 & 39.3\pm8.4 & 41.8\pm3.2 & 32.9\pm0.4 & 70.3\pm1.4 & 57.9\pm0.9 \newline
> \text{+ContextVLA} & \bf{97.9}\pm0.5 & \bf{98.9}\pm0.6 & \bf{96.3}\pm0.3 & \bf{93.1}\pm0.6 & \bf{96.6}\pm0.1 & \bf{53.3}\pm1.8 & \bf{56.0}\pm2.3 & \bf{41.3}\pm2.7 & \bf{74.0}\pm7.2 & \bf{56.2}\pm1.8 & \bf{35.6}\pm0.8 & \bf{70.4}\pm0.2 & \bf{58.8}\pm0.2 \newline
> \hline
> \pi_0\text{-FAST} & 96.3\pm1.1 & 97.5\pm0.9 & 94.5\pm0.6 & 84.8\pm1.4 & 93.3\pm0.5 & 59.0\pm1.2 & 79.0\pm1.2 & 65.0\pm1.9 & 33.0\pm1.0 & 59.0\pm0.7 & 46.1\pm1.0 & 68.1\pm0.8 & 60.3\pm0.1 \newline
> \text{+ContextVLA} & \bf{97.8}\pm0.6 & \bf{98.9}\pm0.4 & \bf{95.9}\pm0.6 & \bf{90.8}\pm0.9 & \bf{95.8}\pm0.2 & \bf{60.7}\pm1.5 & \bf{81.3}\pm5.4 & \bf{78.7}\pm4.3 & \bf{62.0}\pm9.2 & \bf{70.7}\pm1.7 & \bf{48.6}\pm0.6 & \bf{68.7}\pm0.6 & \bf{62.0}\pm0.2 \newline
> \hline
> \text{GR00T N1.5} & 98.0\pm0.5 & \bf{99.3}\pm0.2 & 96.9\pm0.3 & 88.7\pm0.3 & 95.7\pm0.2 & \bf{30.0}\pm1.5 & 28.0\pm1.2 & \bf{16.0}\pm3.5 & 42.7\pm1.8 & 29.2\pm0.9 & 51.8\pm1.4 & 67.6\pm1.1 & 62.3\pm0.3 &  \newline
> \text{+ContextVLA} & \bf{98.6}\pm0.2 & 99.1\pm0.2 & \bf{97.3}\pm0.1 & \bf{93.0}\pm0.3 & \bf{97.0}\pm0.1 & 28.0\pm2.3 & \bf{29.3}\pm0.7 & 14.7\pm3.3 & \bf{50.3}\pm2.4 & \bf{31.8}\pm1.5 & \bf{52.8}\pm0.5 & \bf{70.2}\pm0.5 & \bf{64.4}\pm0.2 \newline
> \hline
> \end{array}

---

> ### Author Response · Authors · 2025-11-21
> **Response to Reviewer 5SdM (3/3)**
>
> **[Q5-1] Analysis on the number of frames.**
>
> **[A5-1]** We would like to clarify that we already reported the ablation on the number of frames in Table 6.
>
> ---
>
> **[Q5-2] Analysis on (1) pooling method, and (2) number of history tokens.**
>
> **[A5-2]** Following your constructive suggestions, we additionally analyze pooling methods and the number of history tokens. We include the results and discussions in the revision (see Appendix E).
>
> First, for the pooling methods, we additionally evaluate Perceiver Resampler [3,4] and Attention Pooling [5] within our framework ($\pi_0$ + ContextVLA). As shown in table below, we observe that when compressing past observations into a single token, global average pooling performs best (56.2% vs. 53.0% for Perceiver Resampler on Simpler-WidowX). In particular, Perceiver Resampler needs 64 tokens to achieve comparable performance to our single-token global pooling, highlighting the efficiency of our method in extracting temporal context with extremely compressed tokens.
>
> Second, for the number of history tokens, we additionally experiment with compressing the past observations into $n$ = 8 or 64 tokens. Specifically, we reshape the flattened tokens of length $N$ into a tensor of shape ($n$, $N/n$), and then apply average pooling along the second dimension to obtain $n$ tokens. As shown in the table below, we observe that compressing historical observations into a single token is the optimal choice, while compressing historical observations into multiple context tokens still consistently outperforms the baseline $\pi_0$ that processes 8-frame observations without compression.
>
> \begin{array}{lcc}
> \hline
> \text{Pooling method} & \\# \text{token} & \text{Simpler-WidowX (\\%)} \newline
> \hline
> \text{Perceiver Resampler} [1,2] & 1 & 53.0 \newline
>                                  & 8 & 53.0 \newline
>                                  & 64 & 56.0 \newline
> \hline
> \text{Attention Pooling} [3]     & 1 & 52.0 \newline
>                                  & 8 & 51.5 \newline
>                                  & 64 & 53.0 \newline
> \hline
> \text{Global Avg. Pooling (Ours)} & 1 & \bf{56.2} \newline
> & 8 & 56.0 \newline
> & 64 & 54.5 \newline
> \hline
> \text{No compression} & 2048 & 47.8 \newline
> \hline
> \end{array}
>
> [1] Alayrac et al., Flamingo: A Visual Language Model for Few-Shot Learning, NeurIPS 2022 \
> [2] Jain et al., Vid2Robot: End-to-End Video-Conditioned Policy Learning with Cross-Attention Transformers, RSS 2024 \
> [3] Ryoo et al., xGen-MM-Vid (BLIP-3-Video): You Only Need 32 Tokens to Represent a Video Even in VLMs, NeurIPS 2024

---

### Official Review · Reviewer_dbTk · 2025-11-01

**Soundness:** 3
**Presentation:** 3
**Contribution:** 2
**Rating:** 6
**Confidence:** 4

**Summary:**

The paper proposes to compress the multi-frame visual observations for VLA models into single context token. The single context token is created by average pooling the hidden states from past observations and action decoder generates action based on this context token concatenated with the current visual observation features. The paper discusses results on simulated and real world tabletop tasks showing how having eight frames in context window instead of single observation helps improve the task success rate.

**Strengths:**

The paper uses a simple general mechanism of summarizing multi-frame visual observation into single context and using KV cache to reduce latency. Instead of prior regression reported with BC policies trained from scratch, the paper shows that VLM-backbone prevents this and performs at par if not better than single observation based policy baselines.
The performance improvement delta across real robotic tasks, which need previous history to succeed, is promising.
The paper shares key insight that VLM initialization enables use of multi-frame inputs as compared to BC policies trained from scratch.

**Weaknesses:**

The paper proposes very simple compression (AvgPool) and doesn't compare to learned pooling (like in Flamingo, Vid2Robot) or other attention summarizers (like Spatio-temporal attentional pooling in BLIP-3-Video).

The evaluation suite of tasks are quite short horizon and seems to favor fixed insertion depth of 2.

The real world eval is limited to tabletop tasks with 50 demos each, unclear if the benefits would transfer where the viewpoint changes significantly between frames, like tasks requiring mobile manipulation, and bimanual dexterity (that often cause occlusion and requires history).

**Questions:**

Can you report the variance and confidence scores along with the success rates? Libero benchmark seems saturated and unclear if the improvement is statistically significant.
Can you highlight why heterogeneous fine-tuning protocols, for example with pi-0 and GR00T N1.5?
While the ablations show more frames help, it is important to note failure cases. Is the policy sensitive to when the context summary is created per task.

---

> ### Author Response · Authors · 2025-11-21
> **Response to Reviewer dbTk (1/2)**
>
> Dear Reviewer dbTk,
>
> We sincerely appreciate your insightful comments and efforts in reviewing our manuscript. We address your comments and questions one-by-one below. In the revised draft, we mark our major revisions as “blue”.
>
> ---
>
> **[W1] Lack of comparison to other compression methods, e.g., learned pooling (Flamingo, Vid2Robot) or attention summarizers (BLIP-3-Video).**
>
> **[A1]** Thank you for pointing this out. We did not consider comparison to different compression methods that you mentioned because our primary focus is more on investigating whether VLAs can leverage multi-frame temporal context through VLM representations themselves, rather than their technical details. However, we fully agree that providing such comparison could provide great insight to readers.
>
> Following your suggestions, we additionally evaluate Perceiver Resampler [1,2] and Attention Pooling [3] within our framework ($\pi_0$ + ContextVLA). As shown in the table below, we observe that when compressing past observations into a single token, global average pooling performs the best (56.2% vs 53.0% for Perceiver Resampler on Simpler-WidowX). In particular, Perceiver Resampler needs 64 tokens to achieve comparable performance to our single-token global pooling,  highlighting the efficiency of our method in extracting temporal context with extremely compressed token. We include the results and discussion in the revision (see Appendix E).
>
> \begin{array}{lcc}
> \hline
> \text{Pooling method} & \\# \text{token} & \text{Simpler-WidowX (\\%)} \newline
> \hline
> \text{Perceiver Resampler} [1,2] & 1 & 53.0 \newline
>                                  & 8 & 53.0 \newline
>                                  & 64 & 56.0 \newline
> \hline
> \text{Attention Pooling} [3]     & 1 & 52.0 \newline
>                                  & 8 & 51.5 \newline
>                                  & 64 & 53.0 \newline
> \hline
> \text{Global Avg. Pooling (Ours)} & 1 & \bf{56.2} \newline
> \hline
> \end{array}
>
> [1] Alayrac et al., Flamingo: A Visual Language Model for Few-Shot Learning, NeurIPS 2022 \
> [2] Jain et al., Vid2Robot: End-to-End Video-Conditioned Policy Learning with Cross-Attention Transformers, RSS 2024 \
> [3] Ryoo et al., xGen-MM-Vid (BLIP-3-Video): You Only Need 32 Tokens to Represent a Video Even in VLMs, NeurIPS 2024
>
> ---
>
> **[W2] Evaluation tasks are quite short-horizon.**
>
> **[A2]** We believe that evaluation already includes relatively long-horizon tasks: Libero-Long requires performing pick-and-place twice, and all of our real-world tasks consist of two sequential sub-tasks. In particular, a real-world CoverNStack task (i.e., covering a cube with a cup, then stacking the cup) requires long-horizon understanding as the policy must remember where the occluded object is located once the cube becomes occluded after the covering.
>
> Moreover, we note that ContextVLA shows larger gains on longer-horizon tasks (e.g., 4 to 5% gains on Libero-Long) compared to shorter-horizon ones (e.g., 1% gains on Libero-Goal). This indicates that ContextVLA becomes more beneficial as the temporal complexity of the task increases.
>
> Nonetheless, to further address the reviewer’s concern, we additionally report the experimental results on the CALVIN (ABC $\rightarrow$ D) benchmark [1], known as the long-horizon benchmark, as each task requires the policy to perform 5 different subtasks sequentially. Specifically, we train both $\pi_0$ and $\pi_0$ + ContextVLA on the training dataset collected from environments A, B, and C for 60K iterations with a batch size of 32. We then evaluate each model on the environment D with 1000 trials and report the average number of successes among 5 sequential subtasks. As shown in the table below, ContextVLA significantly improves $\pi_0$, for example, ContextVLA achieves success rates of 69% in completing all 5 tasks consecutively, whereas $\pi_0$ achieves 60%. This demonstrates the effectiveness of ContextVLA on long-horizon tasks. We report the results in Appendix D in the revision.
>
> \begin{array}{lccccc|c}
> \hline
> \\text{Method} & \\text{1/5} & \\text{2/5}  & \\text{3/5}  & \\text{4/5}  & \\text{5/5}  & \\text{Avg. success length} \newline
> \hline
> \pi_0 & 90.95 & 82.53 & 74.00 & 66.95 & 59.58 & 3.740 \\\\
> \text{+ContextVLA} & \bf{93.38} & \bf{92.70} & \bf{85.80} & \bf{77.58} & \bf{69.36} & \bf{4.238} \\\\
> \hline
> \\end{array}
> [1] Mees et al., CALVIN: A Benchmark for Language-Conditioned Policy Learning for Long-Horizon Robot Manipulation Tasks, RAL 2022

---

> ### Author Response · Authors · 2025-11-21
> **Response to Reviewer dbTk (2/2)**
>
> **[W3] Real-world evaluation: Unclear transferability to tasks beyond the tabletop with significant viewpoint changes between frames, e.g., mobile or bimanual manipulation.**
>
> **[A3]** Our real-world evaluation setup already involves substantial viewpoint changes, as we use a wrist-view camera that moves with the robot arm (see Appendix B.2 for more details). We also expect ContextVLA can perform well on mobile or bimanual manipulation. This is because ContextVLA explicitly captures temporal dependencies (e.g., when objects are occluded or require history) and already shows improved performance under conditions with viewpoint changes compared to existing VLAs.
>
> ---
>
> **[Q4] Improvement significance on Libero benchmark is unclear; Report the variance along with the success rates.**
>
> **[A4]** Following your suggestion, we train each model with 3 different random seeds and report the mean success rate with a standard deviation. As shown in the table below, the improvement is consistent and significant, e.g., on the Libero-Long benchmark, $\pi_0$-FAST gets success rates (%) of 84.8$\pm$1.4, while ContextVLA gets 90.8%$\pm$0.9. We also updated Table 1 accordingly in the revision.
>
> \begin{array}{lccccc}
> \hline
> \text{Method} & \text{Spatial} & \text{Object} & \text{Goal} & \text{Long} & \text{Avg} \newline
> \hline
> \pi_0 & 96.3\pm0.3 & 97.3\pm0.4 & 96.2\pm0.3 & 88.8\pm0.3 & 94.7\pm0.1 \newline
> \text{+ContextVLA} & \textbf{97.9}\pm0.5 & \textbf{98.9}\pm0.6 & \textbf{96.3}\pm0.3 & \textbf{93.1}\pm0.6 & \textbf{96.6}\pm0.1 \newline
> \hline
> \pi_0\text{-FAST} & 96.3\pm1.1 & 97.5\pm0.9 & 94.5\pm0.6 & 84.8\pm1.4 & 93.3\pm0.5 \newline
> \text{+ContextVLA} & \textbf{97.8}\pm0.6 & \textbf{98.9}\pm0.4 & \textbf{95.9}\pm0.6 & \textbf{90.8}\pm0.9 & \textbf{95.8}\pm0.2 \newline
> \hline
> \text{GR00T N1.5} & 98.0\pm0.5 & 99.3\pm0.2 & 96.9\pm0.3 & 88.7\pm0.8 & 95.7\pm0.2 \newline
> \text{+ContextVLA} & \textbf{98.6}\pm0.2 & \textbf{99.1}\pm0.2 & \textbf{97.3}\pm0.1 & \textbf{93.0}\pm0.3 & \textbf{97.0}\pm0.1 \newline
> \hline
> \end{array}
>
> ---
>
> **[Q5] Can you highlight why you consider heterogeneous fine-tuning protocols, e.g., with pi0 and GR00T N1.5?**
>
> **[A5]** We just follow the official implementation and standard fine-tuning protocol of each model, $\pi_0$, $\pi_0$-FAST, and GR00T N1.5. Specifically, for $\pi_0$ and $\pi_0$-FAST, the official implementation recommends fine-tuning all model parameters, but GR00T N1.5 typically freezes the vision encoder and VLM backbones. We include this discussion in the revision (see Appendix A).
>
> ---
>
> **[Q6] Failure case?**
>
> **[A6]** If a single-frame observation already provides sufficient information to perform the task (i.e., Markovian),  ContextVLA shows only marginal performance. For example, several tasks in Libero-Spatial, Object, and Goal are simple, short-horizon pick-and-place with multi-view camera setups, which generally makes them Markovian, and the performance gain becomes quite marginal here.
>
> ---
>
> **[Q7] Is the policy sensitive to when the context summary is created per task?**
>
> **[A7]** Context token is not created per task. Instead, ContextVLA obtains a context token from the latest N frames, regardless of which task the policy is performing.

---

> ### Author Response · Authors · 2025-11-28
> **Gentle Reminder**
>
> Dear Reviewer dbTk,
>
> Thank you again for your time and thoughtful efforts in reviewing our paper.
>
> As the discussion period comes within a week of its end, we would like to gently remind you in case you have any remaining comments. We believe that we have sincerely and successfully addressed your concerns, supported by the corresponding additional experimental results.
>
> If you have any further concerns or questions, please feel free to let us know.
>
> Thank you very much \
> Authors

---

### Author Response · Authors · 2025-11-21
**General Response**

Dear reviewers and AC,

We sincerely appreciate your valuable time and effort spent reviewing our manuscript.

As reviewers highlighted, we address an important problem (5SdM,2Jbw) and our method is a simple yet effective (ALL Reviewers), efficient (dbTk,5SdM,2Jbw), shows promising performance (dbTk,2Jbw) with SOTA performance on diverse benchmarks (XBWu), and provides concrete ablation studies (2Jbw).

We appreciate your constructive comments on our manuscript. In response to the comments, we have carefully revised and enhanced the manuscript with the following additional discussions and experiments:
- Analysis of compression methods and the number of history tokens (Appendix E)
- Visualization of token similarity (Appendix E)
- CALVIN benchmark results (Appendix D)
- Standard deviation along with the success rates (Figure 1 (b), Tables 1, 2, and 3)
- Results of ContextVLA pre-trained on OXE dataset (Tables 1 and 2)
- Related works on the problem of multi-frame observations for BC policies (Appendix H)
- Clarification about experimental setup and benchmark details (Appendix A, B)

In the revised manuscript, these updates are temporarily highlighted in “Blue” for your convenience to check.

We hope our response and revision sincerely address all the reviewers’ concerns.

Thank you very much,\
Authors.

---

### Meta-Review · Area_Chair_54NP · 2026-01-10

**Summary:**

The paper attempts to address a critical bottleneck in Vision-Language-Action (VLA) models: the computational cost of processing multi-frame temporal context. The reviewers raised significant concerns regarding the technical depth, the novelty of the compression strategy, and the robustness of the learned representations. The core concerns including the limited flexibility of the model, and the lack of a deeper theoretical or empirical investigation into the information preserved within the context token.

**Reviewer Concerns:**

Addressed Concerns:
1. Concerns regarding statistical significance, as authors provide mean and standard deviation across multiple seeds.
2. Results from the CALVIN benchmark are included.

Outstanding Concerns:
1. Inference-time robustness. The performance when the number of frames differs from the training setting remains unproven, suggesting the model may be overfitting to a specific input structure rather than learning generalized temporal summaries.
2. Concerns about the context token was only partially addressed with basic similarity matrices, a more rigorous investigation is still needed.

**Reviewer Scores:**

Reviewer XBWu will increase the score from 4 to 5, while other reviewers will not change their scores.

---

### Decision · Program_Chairs · 2026-01-26

Reject